# F-actin disassembly by the oxidoreductase MICAL1 promotes mechano-dependent VWF-GPIbα interaction in platelets

Jean Solarz [1], Christelle Soukaseum[1], Stéphane Frémont[2], Sébastien Eymieux [3,4], Camilia Nabli[3], Christelle Repérant [1], Elisa Rossi [5], Jean-Claude Bordet[6], Cécile V. Denis [1,7], Pierre Mangin [8], Yacine Boulaftali [9], R. Jeroen Pasterkamp [10], Hana Raslova[11], Dominique Baruch[12], Frédéric Adam [1,13], Arnaud Echard [2,13] ✉ & Alexandre Kauskot [1,13] ✉

Mechano-dependent interactions are key to thrombus formation and hemostasis, enabling stable platelet adhesion to injured vessels. The interaction between von Willebrand factor (VWF) and the platelet receptor GPIb-IX-V is central to this process. While GPIbα connects to the actin cytoskeleton, whether actin dynamics are important for GPIbα function under hemodynamic, high shear conditions remains largely unknown. Here, we show that actin disassembly is critical for proper VWF-GPIbα binding under shear. Mechanistically, we identify the oxidoreductase MICAL1 as a shear-activated regulator that promotes local F-actin disassembly around the GPIb-IX-V complex. This enables its translocation to lipid rafts and reinforces VWF binding. MICAL1-deficient platelets display impaired adhesion, increased deformability under shear, and defective thrombus formation in vivo. Thus, MICAL1 drives shear-dependent actin remodeling that supports GPIb-IX-V mechanotransduction and platelet function. These findings uncover a role for actin oxidation in platelet adhesion, providing a connection between cytoskeletal redox control and platelet function during thrombus formation.

The ability of platelets to adhere to sites of vascular injury is essential for the arrest of bleeding and subsequent vascular repair. Von Willebrand factor (VWF) - a large multimeric protein found in plasma - is a key adhesive protein that initiates platelet-vessel wall interactions under high shear stress[1]. Once immobilized on collagen fibers that become accessible at the site of vessel wall injury, the A1 domain of VWF captures rolling platelets from rapidly flowing blood via specific interactions with the platelet glycoprotein (GP) Ibα/β-IX-V complex. Using optical tweezers to exert pulling forces on the isolated A1 domain that binds to the N-terminal part of GPIbα, the unfolding of a juxta-membrane domain —the mechanosensory domain (MSD)— was observed upon continuous pulling force[2,3]. Although distant from the N-terminal region of GPIbα, unfolding of the MSD enhances the lifetime of the interaction between GPIbα and the A1 domain of vWF[4].

[1]HITh, UMR_S1176, INSERM, Université Paris-Saclay, Le Kremlin-Bicêtre, France. [2]Institut Pasteur, Université Paris Cité, CNRS UMR3691, Paris, France. [3]Microscopy facility, US61 ASB, University of Tours, University Hospital Center of Tours, Inserm, Tours, France. [4]INSERM U1259 MAVIVHe, University of Tours, Tours, France. [5]Université Paris-Cité, INSERM, Optimisation thérapeutique en neuropharmacologie, OTEN U1144, Paris, France. [6]UR4609 Hémostase & Thrombose, Université Claude Bernard Lyon, Lyon, France. [7]CHRU Nancy, Vandœuvre-lès-Nancy, France. [8]INSERM U1255, Université de Strasbourg, Strasbourg, France. [9]INSERM U1148, Université Paris Cité, Paris, France. [10]Department of Translational Neuroscience, University Medical Center Utrecht, Brain Center, Utrecht University, Utrecht, The Netherlands. [11]INSERM U1287, Institut Gustave Roussy, Université Paris Saclay, Villejuif, France. [12]INSERM U1140, Université Paris Cité, Paris, France. [13]These authors contributed equally: Frédéric Adam, Arnaud Echard, Alexandre Kauskot. ✉e-mail: arnaud.echard@pasteur.fr; alexandre.kauskot@inserm.fr

However, this is not sufficient to immobilize platelets at the site of injury, and additional mechanisms are required to withstand the mechanical forces generated by the blood flow. These include the extension of plasma membrane tethers filled with actin filaments that promote platelet capture, which contribute to firm adhesion and spreading[5,6]. Stable platelet adhesion to VWF also requires the activation of the integrin αIIbβ3, which binds directly to VWF in the RGD peptide motif in the C4 domain[7,8]. Following adhesion to the site of injury, immobilized platelets aggregate one to another through GPIb-IX-V binding to VWF and αIIbβ3 binding to fibrinogen[9], leading to the formation of a thrombus anchored to the damaged vessel, which occludes the vascular breach and stops bleeding.

VWF engagement with GPIbα under physiological shear stress induces MSD unfolding on platelets and propagates a signal across the plasma membrane to the cytoplasmic domain of GPIbα[2]. This signal is also dependent on the VWF-induced receptor clustering, which depends on GPIb-IX-V translocation to particular lipid domains of the plasma membrane also referred as lipid rafts. These lipid domains are necessary for functional GPIb regulation[10], distribution[11] and platelet adhesion[12–15]. Ultimately, VWF-GPIbα interactions under shear stress result in intracellular signaling leading to tether extension and integrin αIIbβ3 activation to promote stable platelet adhesion. Interestingly, initial VWF-GPIbα interactions stimulate the global assembly of actin filaments (F-actin) in platelets[16] and the actin-binding protein Filamin A - which binds directly to the cytoplasmic domain of GPIbα - participates to the GPIb-IX-V signaling pathway[17,18]. Mechanistically, Filamin A may limit the lateral mobility of the GPIb-IX-V complex in the plane of the plasma membrane, thereby influencing the number of active bonds formed between GPIbα and VWF[13]. In addition, it has been described that actin-dependent cytoskeletal forces are transmitted to VWF through the GPIb-IX-V complex to reinforce VWF-GPIbα interactions[19]. Finally, global actin depolymerization using drugs leads to enhanced aggregation when VWF interacts with GPIbα in the absence of shear stress and also influences platelet tether size upon shear stress[5]. However, the role of actin polymerization and dynamics in GPIb-IX-V signaling under shear stress remains poorly understood. Open questions include whether actin polymerization/depolymerization plays a functional role in stable platelet adhesion at the site of injury, how actin dynamics is regulated and whether this is important for platelet signaling induced by VWF-GPIbα interactions under hemodynamic conditions.

Here, we found that actin disassembly in platelets, specifically under high shear stress, is critical for proper VWF-GPIbα interactions to promote stable platelet adhesion. We show that actin disassembly in the GPIb-IX-V complex relies on the recruitment of MICAL1 (molecule interacting with CasL protein 1) to the GPIb-IX-V complex shortly after platelet activation. MICAL1 belongs to a conserved family of flavoenzyme oxidoreductases called MICAL, which directly oxidize two specific methionine residues in actin filaments, leading to their rapid disassembly[20–31]. These enzymes regulate key cellular functions, including neuronal axon guidance, cell migration, cytokinesis and viral budding[30,32–43], but their potential role in hemostasis was unknown. Mechanistically, we show that MICAL1 promotes the translocation of the GPIb-IX-V complex into lipid rafts by limiting F-actin polymerization and counteracts shear-induced morphological changes, both of which contribute to stabilizing platelet adhesion in vitro and in vivo. Altogether, this work demonstrate that actin polymerization must be restrained for proper platelet adhesion at the site of injury and reveals a key role for post-translational oxidation of actin in platelet function.

## Results

### Platelet-VWF interaction under high shear requires F-actin disassembly to promote efficient platelet adhesion and stability

To investigate the role of actin dynamics in platelet-VWF interaction under shear, we treated mouse whole blood with either the F-actin depolymerizing drug Latrunculin-A (LatA) or the F-actin stabilizing drug Jasplakinolide (Jasp), and measured platelet adhesion on VWF matrix at arteriolar shear rate of $1500 \, \text{s}^{-1}$ and stable platelet adhesion at $9000 \, \text{s}^{-1}$ (Supplementary Fig. 1a, b). As expected, LatA and Jasp respectively decreased and increased platelet actin in the cytoskeletal, Triton X-100 detergent-resistant fraction (Supplementary Fig. 1c). We observed that F-actin depolymerization by LatA enhanced platelet adhesion without affecting platelet stability (Fig. 1a, b). In contrast, F-actin stabilization by Jasp strongly inhibited both adhesion and stability (Fig. 1a, b). Since platelet adhesion to VWF depends on both VWF-GPIbα and on VWF-αIIbβ3 integrin interactions, we specifically investigated the contribution of VWF-GPIbα interaction by measuring platelet rolling velocities at an arteriolar shear rate of $1500 \, \text{s}^{-1}$, after αIIbβ3 integrin blockade. We observed that F-actin depolymerization decreased rolling velocities, whereas F-actin stabilization increased them (Fig. 1c, Supplementary Movies 1–3). Thus, F-actin depolymerization favors VWF-GPIbα interactions, whereas F-actin stabilization is detrimental for these interactions under high shear. In contrast, in the absence shear (static condition), F-actin stabilization did not alter VWF binding to platelets (Fig. 1d). This suggests that mechano-dependent binding of GPIbα with associated proteins and/or signaling downstream of the VWF-GPIbα interaction important for platelet adhesion is impaired upon F-actin stabilization. Consistently, co-immunoprecipitation of GPIbα revealed that LatA treatment decreased F-actin association to the receptor, whereas Jasp treatment increased this association (Fig. 1e, f). These findings suggest that mechanical forces regulate VWF-GPIbα interactions under high shear conditions. Altogether, we conclude that platelet interaction with VWF under shear requires F-actin disassembly to promote efficient platelet rolling, adhesion and stability.

### F-actin induces the recruitment of MICAL1 to the GPIb-IX-V complex upon platelet activation

We next sought to identify the mechanism by which F-actin disassembly is achieved to maintain optimal VWF-GPIbα interaction. Analysis of the Biological General Repository for Interaction Datasets (BioGRID[4,4])[44] database, revealed that GPIbα and GPIbβ interacted − directly or indirectly− with 81 different proteins (Supplementary Data 2). Of these, only MICAL1 is known to control F-actin depolymerization (Fig. 2a and Supplementary Data 2). MICAL1 is an enzyme that catalyzes the oxidation of actin filaments, leading to their severing and disassembly[27–30,45,46] and is expressed in mouse and human platelets[47,48]. We thus postulated that MICAL1 could regulate VWF-GPIbα-dependent adhesion under high shear by controlling F-actin disassembly upon platelet activation.

To test this hypothesis, we first analyzed whether there was an interaction between MICAL1, GPIbα and F-actin, given the fact that platelet activation rapidly increases actin polymerization[49]. We found that MICAL1 and GPIbα massively translocated to the Triton X-100-insoluble fraction - which is enriched in crosslinked actin filaments[50] - within 2 min of VWF/ristocetin agonist stimulation (Fig. 2b) (human platelets were used for this experiment because the available MICAL1 antibodies were not sensitive enough to detect the mouse MICAL1 protein). Furthermore, the redistribution of MICAL1 and GPIbα from the soluble to the insoluble cytoskeletal fraction was strongly diminished upon F-actin depolymerization by LatA (Fig. 2b). This indicates that the presence of MICAL1 and GPIbα in the insoluble fraction is largely dependent on their association with F-actin.

We next analyzed whether MICAL1 associates with GPIbα upon platelet activation. Upon co-immunoprecipitation of GPIbα, we found low amounts of both MICAL1 and β-actin associated with GPIbα in resting platelets (Fig. 2c). In contrast, after activation with VWF/ristocetin, a significant recruitment of MICAL1 (fold increase: $17 \pm 3$) and β-actin (fold increase: $20 \pm 2$) was observed (Fig. 2c). These findings led us to investigate whether MICAL1 association with GPIbα is dependent on actin polymerization. Upon F-actin depolymerization using LatA,

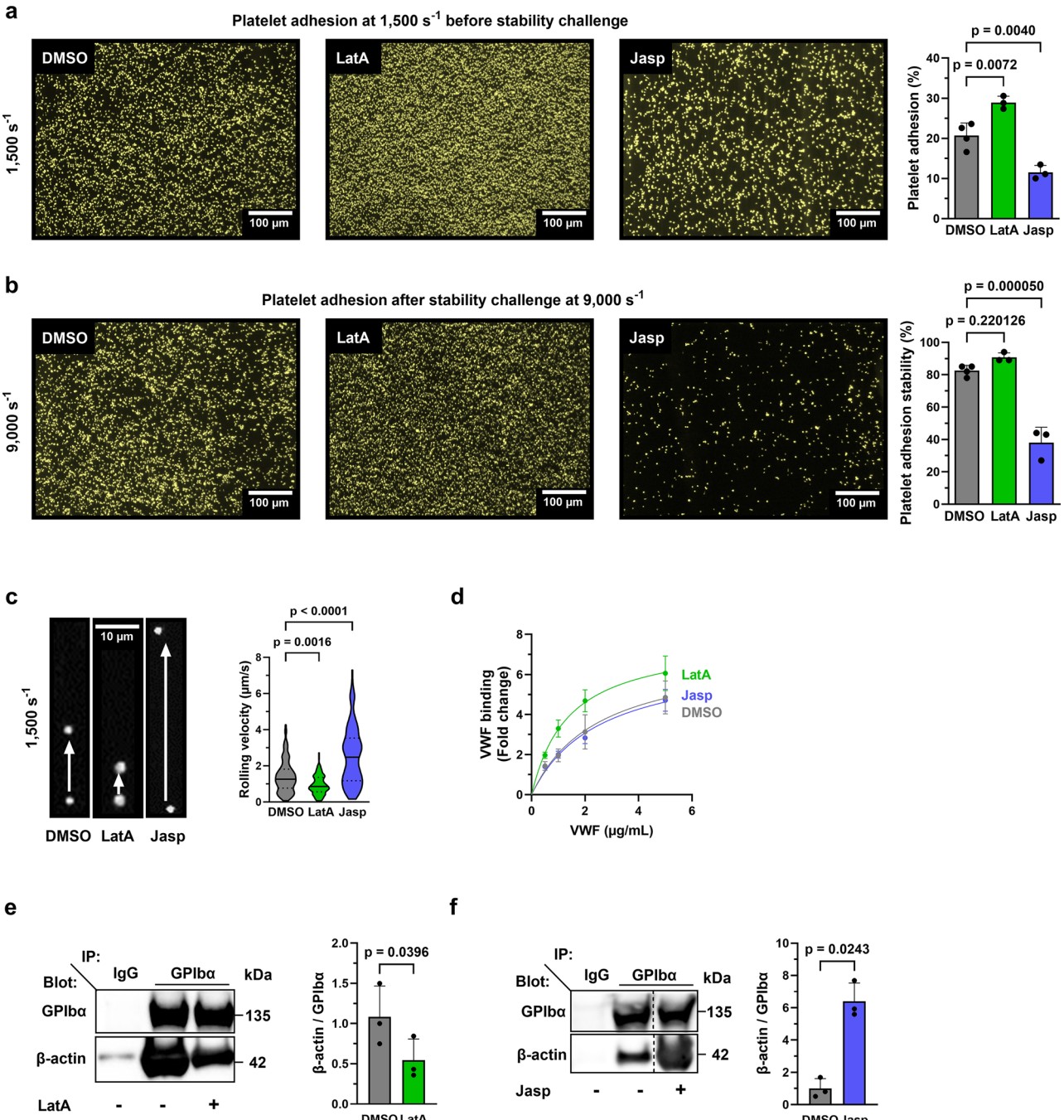

**Fig. 1 | Platelet-VWF interaction under shear requires F-actin disassembly to promote efficient platelet adhesion and stability. a–c** Mouse blood was treated with LatA (500 nM) or Jasp (1 μM) or DMSO before perfusion on recombinant mouse VWF (Supplementary Fig. 1). **a** Platelet adhesion (% of surface coverage) (mean ± SD, N: DMSO = 4, LatA = 3, Jasp = 3 independent experiments, one-way ANOVA with Tukey post-hoc test, F = 39.05, degree of freedom (df) = 7) and **b** platelets adhesion stability (mean ± SD, N: DMSO = 4, LatA = 3, Jasp = 3 independent experiments, one-way ANOVA with Tukey post-hoc test, F = 74.73, df = 7). Left panel: images showing platelet adhesion before (**a**) and after (**b**) the stability challenge. Scale bars = 100 μm. Platelet adhesion and stability were calculated as described in Supplementary Fig. 1. **c** Mouse platelet rolling velocity. Violin plots with the median represented by a central line and the interquartile range (25th–75th percentiles) indicated by the upper and lower lines. (N = 3 independent experiments DMSO = 133, LatA = 131, Jasp = 129 platelets; one-way ANOVA with

Tukey post-hoc test: F = 75.84, df = 390). Supplementary Movies 1, 2 and 3. Left panel: images of platelet velocity with a time-lapse of 10 s. Scale bar = 10 μm. **d** Recombinant mouse VWF binding measured by flow cytometry. Washed mouse platelets were treated with either LatA (500 nM), Jasp (1 μM), or DMSO and activated with botrocetin (5 μg/mL) and a range of mouse VWF. (mean ± SD, N = 3 independent experiments). GPIbα immunoprecipitation (in static conditions) treated with either DMSO or LatA (500 nM) (**e**) or Jasp (1 μM) (**f**). Samples were blotted for GPIbα and β-actin. **e** Mean ± SD, N = 3 independent experiments, two-tailed paired Student's t test, t = 6.297, df = 2. Left panel: representative western blots. **f** Mean ± SD, N = 3 independent experiments, two-tailed paired Student's t test, t = 4.874, df = 2. Left panel: representative western blots. Vertical lines indicate a repositioned gel lane. In histograms, each symbol represents 1 individual. Independent experiments correspond to different mice. Source data are provided as a Source Data file.

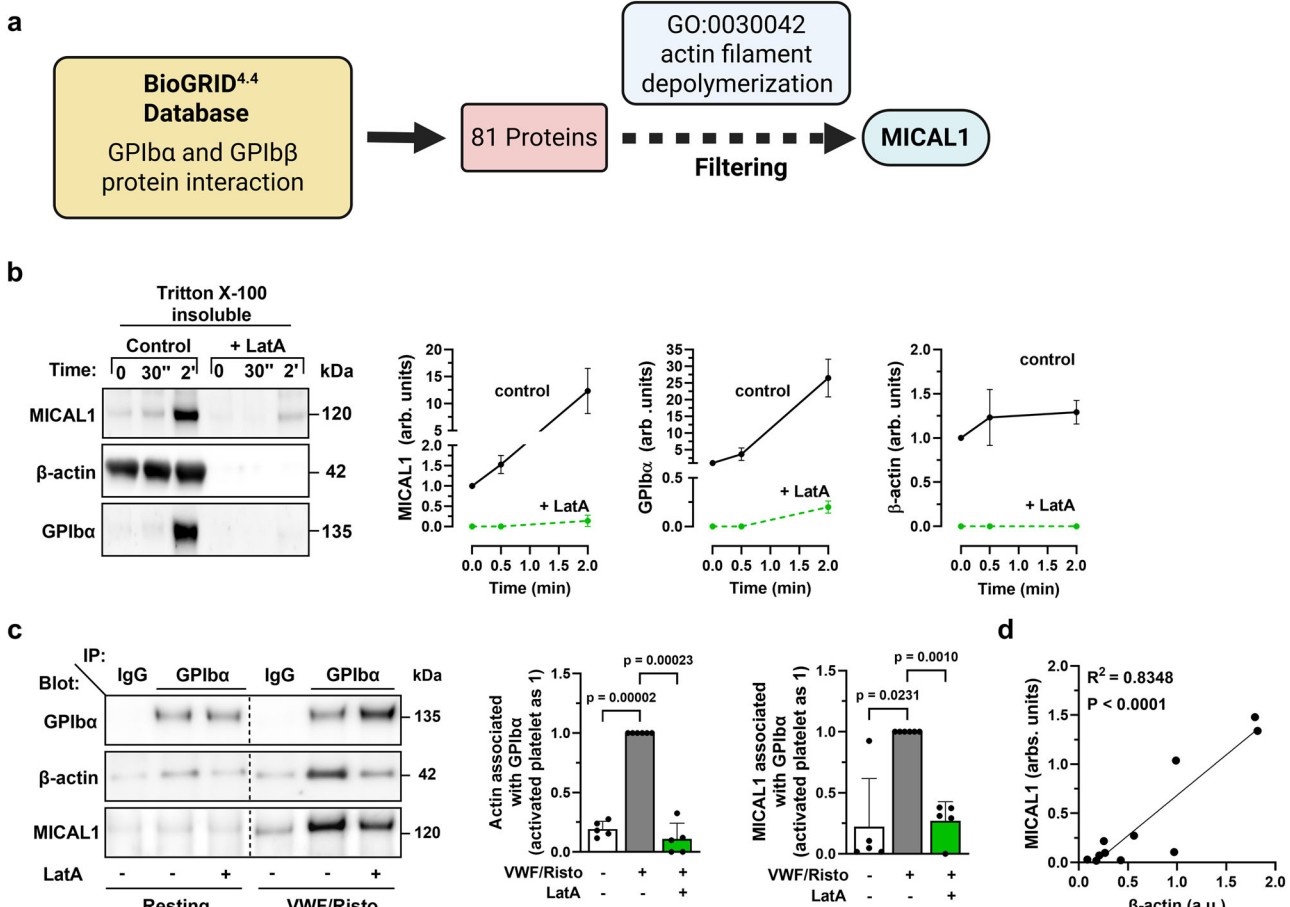

**Fig. 2 | F-actin induces the recruitment of MICAL1 to the GPIb-IX-V complex upon platelet activation. a** Identification of MICAL1 as a potential actin-depolymerizing protein interacting with GPIbα/GPIbβ thanks to the database BioGRID[4.4] and the Gene Ontology "Actin filament depolymerization" (GO: 0030042). Created with BioRender.com. **b** Lysates of human platelets treated or not with LatA (15 μM) at 0 s (resting platelets), after 30 s or 2 min of activation (VWF 10 mg/mL/Risto 0.4 mg/mL) were separated into Triton X-100 soluble and insoluble fractions and blotted. Quantification of insoluble MICAL1 (mean ± SD, $N = 4$ independent experiments from 4 different donors, two-way ANOVA with Šídák post hoc test, $F_{(2, 18)} = 27.06$, 2 min: $p = 0.00000003$), GPIbα (mean ± SD, $N = 3$ independent experiments from 4 different donors, two-way ANOVA, $F_{(2, 12)} = 49.40$, 2 min: $p = 0.00000005$) and β-actin (mean ± SD, $N = 4$ independent experiments from 4 different donors, two-way ANOVA with Šídák post hoc test, $F_{(2, 18)} = 2.424$,

2 min: $p = 0.0000000004$). Left panel: representative western blots. Right panels: MICAL1, GPIbα, β-actin quantification from western blots. **c** GPIbα immunoprecipitation from lysates from washed human platelets in resting or activated conditions (VWF 10 mg/mL/Risto 0.4 mg/mL) human platelets treated with LatA (15 μM) or DMSO as control and blotted for GPIbα, β-actin, (mean ± SD, $N = 5$ independent experiments from 4 different donors, one-way ANOVA with Šídák post hoc test, $F = 202$, df = 4) and MICAL1 (mean ± SD, $N = 5$ independent experiments from 4 different donors, one-way ANOVA with Šídák post hoc test, $F = 19.2$, df = 4). Left Panel: representative western blot. Right panel: β-actin and MICAL1 quantification from western blots. **d** Correlation of the presence of MICAL1 and β-actin associated to GPIbα after immunoprecipitation from experiments performed in (**c**). $R^2$ and $p$ were obtained through simple linear regression. Vertical dotted lines indicate a repositioned gel lane. Source data are provided as a Source Data file.

we observed a marked reduction in both β-actin and MICAL1 within GPIbα after stimulation (Fig. 2c). Remarkably, the presence of MICAL1 in the GPIb-IX-V complex correlated with the presence of β-actin in the complex (Fig. 2d). We conclude that actin polymerization upon VWF/ristocetin activation induces a marked recruitment of MICAL1 to GPIbα.

Altogether, these results reveal that VWF-induced actin polymerization in the initial phase of platelet activation is pivotal for the recruitment of MICAL1 into the GPIb-IX-V complex.

## MICAL1 contributes to hemostasis and thrombosis under high shear

To investigate the role of MICAL1 in platelet function in vivo, we generated tissue-specific knockout mice for MICAL1 in megakaryocyte and platelet lineage by crossing $Mical1^{fl/fl}$ mice with mice carrying Cre recombinase under the control of the $Pf4$ promoter to generate MICAL1 wild-type ($Mical1^{+/+}$) and knock-out ($Mical1^{-/-}$) mice. As

expected, the MICAL1 protein was absent from platelets (Fig. 3a). $Mical1^{-/-}$ mice were viable and born at expected mendelian ratios (Supplementary Table 1) and showed no apparent spontaneous bleeding or thrombosis events from birth to two years of age. Furthermore, $Mical1^{-/-}$ mice showed no overt abnormalities in platelet, red and white blood cell counts (Supplementary Table 2). Of note, transmission electron microscopy (TEM) analysis revealed a slight increase in platelet size in only a subset of the population (12% of platelets) (Fig. 3b–f). To further characterize platelet morphology, we quantified both the major and minor diameters of individual platelets and found that both parameters were significantly increased in $Mical1^{-/-}$ platelets compared to controls, indicating a moderate but consistent increase in platelet size. However, the ratio between the major and minor axes remained unchanged, suggesting that platelet shape was preserved and that platelets in $Mical1^{-/-}$ mice are slightly larger overall (Fig. 3c–e). Consistent with this, the mean platelet volume (MPV) measured by hematology analyzer was not significantly different in $Mical1^{-/-}$

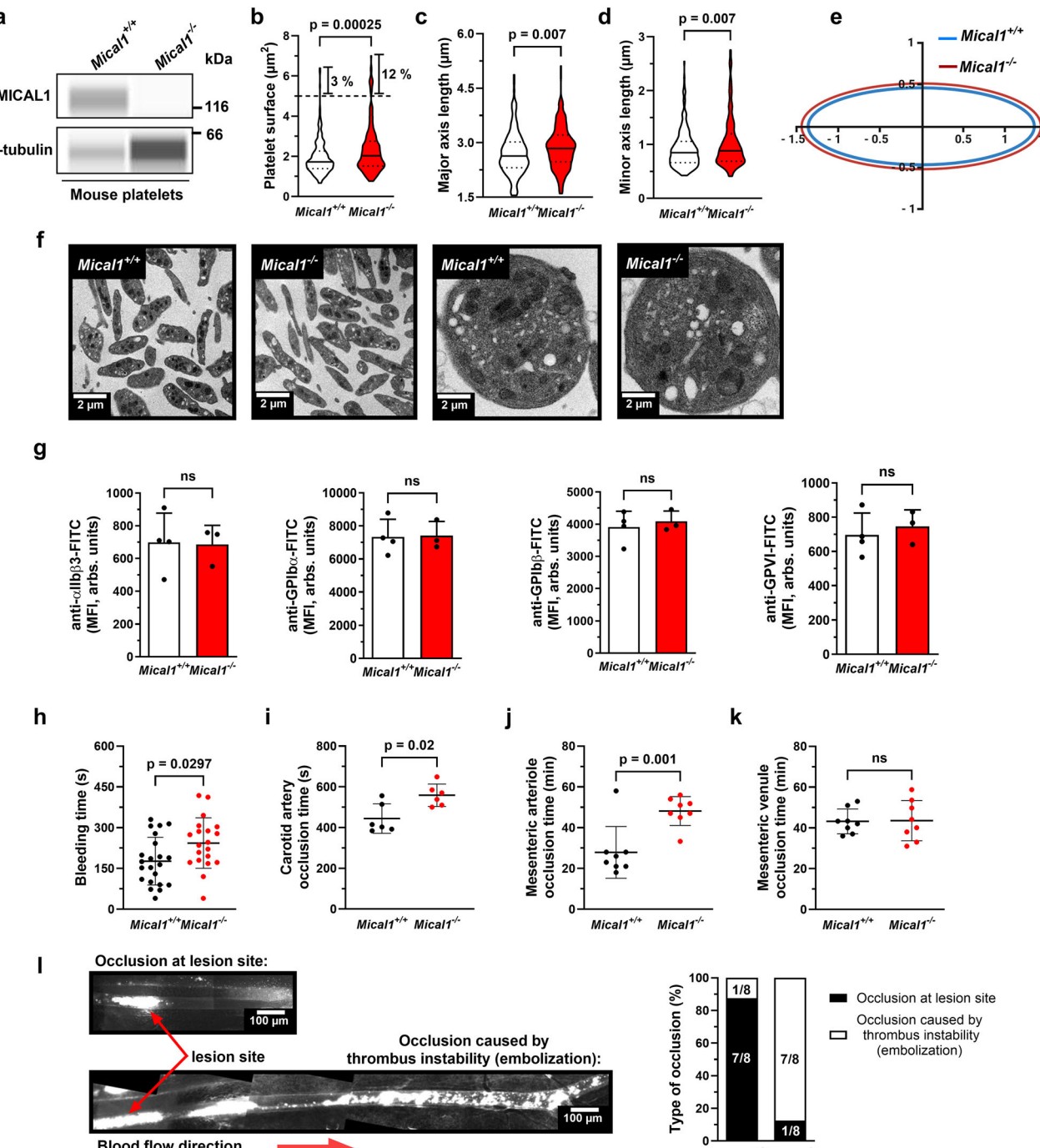

**Fig. 3 | MICAL1 contributes to hemostasis and thrombosis under high shear.**
**a** Mical1 expression in *Mical1*[+/+] and *Mical1*[-/-] platelets analyzed by Wes (α-tubulin as loading control). Source data are provided. **b–e** Platelet ultrastructure assessed by transmission electron microscopy in two independent experiments. **b** Platelet surface area, **c** major axis, and **d** minor axis were measured ($N = 200$ platelets per genotype). Violin plots show median (central line) and interquartile range (25th–75th percentiles, upper and lower lines). Two-tailed unpaired Student's $t$ tests: **b** $t = 3.693$, df = 398; **c** $t = 2.706$, df = 398; **d** $t = 2.699$, df = 398. **e** Schematic of platelet axes (Created with BioRender.com). **f** Representative images of *Mical1*[+/+] and *Mical1*[-/-] platelets. Scale bars: 2 µm (top), 0.2 µm (bottom). **g** Surface expression of αIIbβ3, GPIbα, GPIbβ, and GPVI by flow cytometry (mean fluorescence intensity ± SD; *Mical1*[+/+]: $N = 4$; *Mical1*[-/-]: $N = 3$). Two-tailed unpaired Student's $t$ tests: αIIbβ3, $t = 0.1012$; GPIbα, $t = 0.09656$; GPIbβ, $t = 0.5239$; GPVI, $t = 0.5602$; df = 5. **h** Tail bleeding time (mean ± SD, $N = 22$); two-tailed Mann–Whitney test: U = 134. **i** Time to carotid artery occlusion after FeCl₃ injury (mean ± SD, $N = 6$); paired Student's $t$ test: $t = 3.429$, df = 5. Time to occlusion in mesenteric arterioles (**j**) and venules (**k**) following FeCl₃ injury (mean ± SD, $N = 8$); unpaired t-tests: (**j**) $t = 3.947$, (**k**) $t = 0.08440$; df = 14, $p = 0.93$. **l** Type of occlusion in mesenteric arterioles (lesion site vs. distant embolic occlusion). Left: representative reconstructions. Scale bars: 100 µm. Right: percentage distribution ($N = 8$). ns not significant.

platelets (Supplementary Table 2), likely due to the magnitude of the size increase and limitations in resolution of the automated measurements (Supplementary Table 2). Moreover, the surface expression of major platelet receptors, including GPIbα, was normal (Fig. 3g). Other parameters such as platelet clearance and the number of megakaryocytes in the bone marrow were also normal (Supplementary Fig. 2). We conclude that absence of MICAL1 leads to slightly larger platelets without significant alteration in shape, and that platelet count, clearance and receptors expression remain normal in $Mical1^{-/-}$ mice.

We then investigated the impact of MICAL1 deficiency in vivo on both the hemostatic and prothrombotic functions of platelets by measuring the bleeding times and the thrombus formation (corresponding to both platelet adhesion and aggregation), respectively. $Mical1^{-/-}$ mice showed a moderately prolonged bleeding time in a tail clip assay, compared to $Mical1^{+/+}$ mice ($Mical1^{-/-}$: $244 \pm 21$ s vs. $Mical1^{+/+}$: $177 \pm 19$ s; Fig. 3h). We next assessed thrombus formation following $FeCl_3$-induced injury in different vessels with different shear rates: mesenteric venules (low shear rates; $100$ s$^{-1}$)[51], carotid artery (moderate shear rates: $950$ s$^{-1}$)[52] and mesenteric arterioles (high shear rates: $1400$ s$^{-1}$)[53]. In $Mical1^{-/-}$ mice, carotid occlusion was significantly delayed compared to littermate controls ($Mical1^{-/-}$: $693 \pm 132$ s vs. $Mical1^{+/+}$: $464 \pm 44$ s) (Fig. 3i). Similarly, the time to arterial occlusion was prolonged in the mesenteric arteries of $Mical1^{-/-}$ mice ($Mical1^{-/-}$: $48 \pm 8$ min vs. $Mical1^{+/+}$: $24 \pm 4$ min) (Fig. 3j), whereas no significant differences were observed in the venules (Fig. 3k). In addition, thrombi in $Mical1^{+/+}$ mice remained stable and localized to the injury site in arterioles. In contrast, $Mical1^{-/-}$ mice formed unstable thrombi in arterioles, leading to thromboembolism and subsequent occlusion downstream of the initial lesion (Fig. 3l).

We conclude that MICAL1 is required for a normal hemostatic function and a proper response to thrombotic injury in vivo, presumably by controlling thrombus stability under the high shear stress present in arteries and arterioles.

## MICAL1 promotes thrombi stability under high shear

To elucidate the mechanisms responsible for the impaired thrombus occlusion and thromboembolism observed in MICAL1-deficient mice under high shear, we investigated ex vivo platelet thrombus dynamics by perfusing anticoagulated whole blood over immobilized type I fibrillar collagen (which captures VWF from plasma and mimics injured blood vessels) at an arteriolar shear rate of $1500$ s$^{-1}$. Remarkably, platelet thrombus formation was significantly altered in the absence of MICAL1 compared to controls. In particular, individual thrombi were larger in the absence of MICAL1, as evidenced by increased thrombus volume (Fig. 4a), likely due to increased dense granule secretion since thrombus size was normalized after inhibiting second-wave mediators (ADP/ATP by apyrase) (Supplementary Fig. 3a). Similarly, platelet aggregation and ATP secretion were significantly enhanced in $Mical1^{-/-}$ platelets induced by Thrombin, Cvx, PAR4-AP agonists leading to secretion (strong agonists) but not by ADP (weak agonist) (Supplementary Fig. 3b). In line with the observed ex vivo thrombus formation, treatment with apyrase normalized the aggregation of $Mical1^{-/-}$ platelets to the level observed $Mical1^{+/+}$ platelets. These results indicate that the aggregation enhancement is primarily due to increased release of dense granules (Supplementary Fig. 3c, d). However, in suspension and static conditions (i-e in the absence of platelet aggregation), integrin αIIbβ3 activation and P-selectin exposure (a secretion marker of α-granules) were comparable between $Mical1^{-/-}$ and $Mical1^{+/+}$ mice (Supplementary Fig. 3e, f). Collectively, the formation of larger thrombi at a shear rate of $1500$ s$^{-1}$ and increased aggregation, together with the absence of change in integrin activation in MICAL1 deficient platelets cannot explain the defects in arterial occlusion observed in $Mical1^{-/-}$ mice in vivo.

In contrast, our ex vivo data revealed an increased propensity of thrombi to embolize at high shear rates after stability challenge at $9000$ s$^{-1}$. Quantitative analysis showed that $49 \pm 3\%$ of the thrombus volume embolized within 3 min after buffer perfusion in $Mical1^{-/-}$ samples, compared to $33 \pm 2\%$ in controls (Fig. 4b). We therefore hypothesized that this embolism could be due to impaired adhesion in high shear conditions, reflecting defective interactions between MICAL1-deficient platelets with collagen fibers through GPVI receptors and with the VWF bound to collagen through GPIbα receptors. To specifically measure adhesion, platelet aggregates were prevented by blocking the αIIbβ3 integrin with the Leo.H4 antibody[54]. Adhesion under high shear was significantly impaired in $Mical1^{-/-}$ mice, as evidenced by a $59 \pm 11\%$ reduction of the surface covered by platelets compared to $Mical1^{+/+}$ controls (Fig. 4c).

We conclude that MICAL1 promotes platelet thrombus stability under high shear, in part by controlling platelet adhesion. This likely explains the defects in thrombus stability and subsequent occlusion under high stress observed in arteries and arterioles in $Mical1^{-/-}$ mice.

## Control of F-actin disassembly by MICAL1 supports shear-dependent platelet adhesion

Thrombus formation at high shear relies first on VWF-GPIbα interactions and then, on αIIbβ3 integrin engagement with VWF[8]. We thus investigated the role of MICAL1 in platelet interaction with immobilized VWF at a shear rate of $1500$ s$^{-1}$, followed by a stability challenge at $9000$ s$^{-1}$. Under the experimental conditions tested (no collagen), blood perfusion on immobilized VWF results in platelet adhesion without the formation of aggregates, as previously described[8,9]. $Mical1^{-/-}$ platelets showed significantly reduced platelet adhesion compared to controls at a shear rate of $1500$ s$^{-1}$ (area covered: $14 \pm 1\%$ vs. $20 \pm 1\%$; Fig. 5a). After increasing the shear rate to $9000$ s$^{-1}$, platelet adhesion decreased further ($7 \pm 1\%$ in $Mical1^{-/-}$ vs. $15 \pm 1\%$ in $Mical1^{+/+}$), corresponding to a stability of $53 \pm 3\%$ in $Mical1^{-/-}$ platelets vs. $76 \pm 3\%$ in controls (Fig. 5b). In contrast, no difference in platelet adhesion was observed to VWF in the absence hemodynamics conditions (static condition; Supplementary Fig. 4a). Since stable adhesion relies on the interaction of αIIbβ3 integrin with the RGD sequence in VWF following the engagement of GPIbα, we assessed integrin activation of adherent platelet at a shear rate of $1500$ s$^{-1}$. As measured by the JON/A antibody, αIIbβ3 integrin activation was decreased by 28% in $Mical1^{-/-}$ platelets compared to $Mical1^{+/+}$ platelets at a shear rate of $1500$ s$^{-1}$ (Fig. 5c), but was unchanged in suspension without shear (Supplementary Fig. 4b). Importantly, platelet adhesion and stability to fibrinogen under blood flow conditions, which are exclusively dependent on interaction with αIIbβ3 integrin (and are GPIbα independent), was unchanged in $Mical1^{-/-}$ platelets (Supplementary Fig. 4c), showing that MICAL1 is not directly coupled to αIIbβ3 integrin regulation. These results suggest that MICAL1 promotes platelet adhesion only under shear by regulating the GPIbα pathway.

We next investigated whether MICAL1 controls adhesion by favoring F-actin disassembly, since MICAL family enzymes are known to directly oxidize and disassemble actin filaments. Consistent with this hypothesis, we found that F-actin levels were increased in $Mical1^{-/-}$ platelets compared to controls at a shear rate of $1500$ s$^{-1}$ (Fig. 5d). In contrast, F-actin levels either in resting platelets or under various static conditions in thrombin-activated platelets in suspension or in static adhesion (on fibrinogen or VWF) remained unchanged in $Mical1^{-/-}$ platelets, compared to $Mical1^{+/+}$ platelets (Supplementary Fig. 5a–c). As GPIbα shedding can reduce platelet responses to VWF and may be influenced by actin polymerization[55], we quantified GPIbα expression on adherent platelets to VWF matrix at a shear rate of $1500$ s$^{-1}$. No differences were detected between $Mical1^{-/-}$ and $Mical1^{+/+}$ platelets (Fig. 5e), indicating that the differences in adhesion under-flow of $Mical1^{-/-}$ platelets are not due to GPIbα shedding. Moreover, the expression of actin-binding proteins, myosins, small GTPases, Ras-

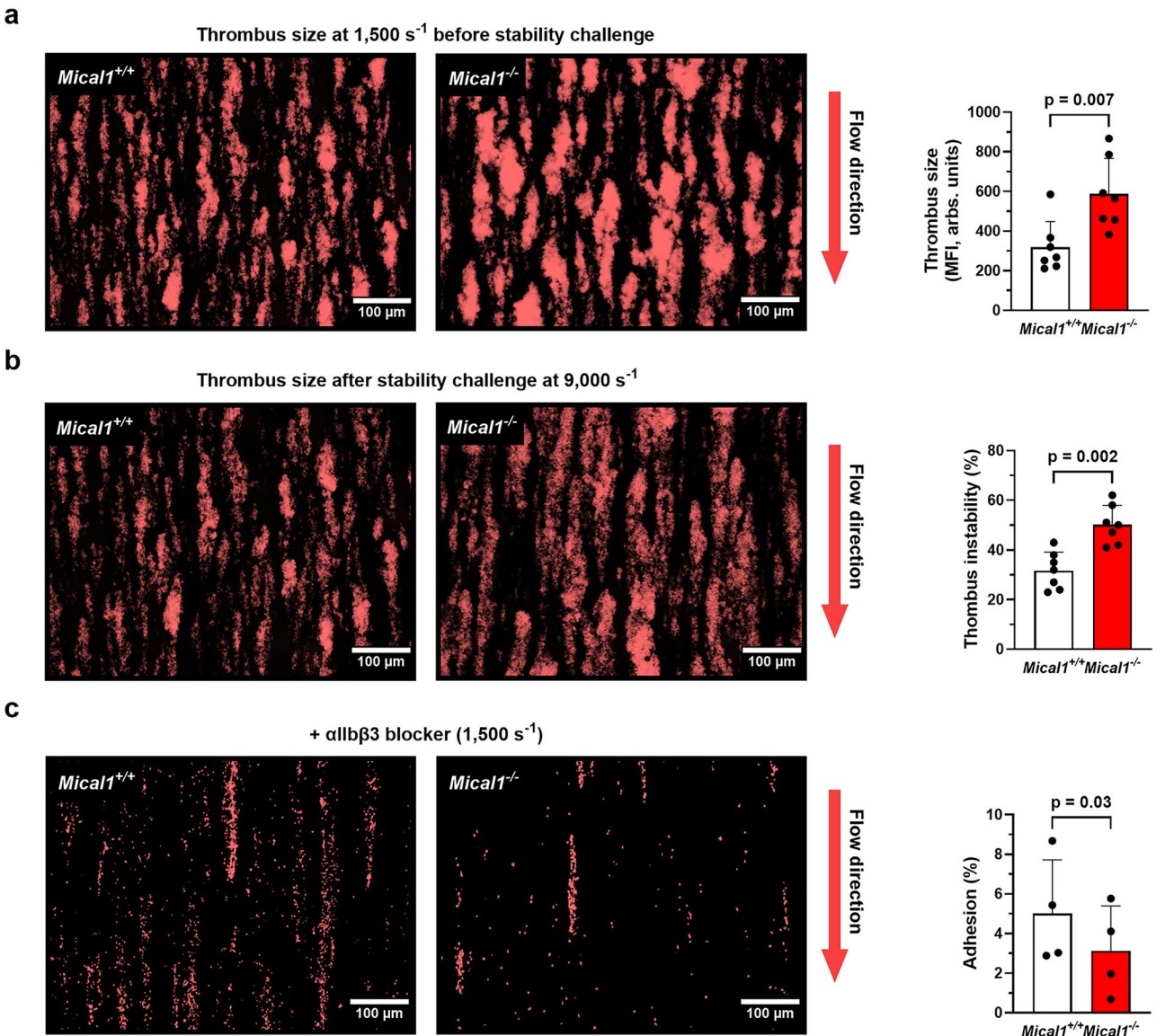

**Fig. 4 | MICAL1 is required for thrombi stability under high shear.**
**a−c** Rhodamine 6G stained *Mical1*⁺/⁺ or *Mical1*⁻/⁻ platelets in whole blood were perfused on type I collagen matrix. **a** Thrombus size at a shear rate of 1500 s⁻¹ (mean fluorescence intensity (MFI) ± SD, $N = 7$ independent experiments; two-tailed unpaired Student's *t* test, $t = 3.227$, df = 12), and **b** the percentage of thrombus instability at a shear rate of 9000 s⁻¹ (see experimental protocol overview in Supplementary Fig. 1) (mean ± SD, $N = 7$ independent experiments; two-tailed Mann−Whitney, U = 2). Left panel: representative images showing **a** thrombus size before and **b** after stability challenge. **c** Platelet adhesion in the absence of thrombus formation was assessed using αIIbβ3 blocking antibody (Leo.H4) (mean ± SD, $N = 4$ independent experiments; two-tailed paired Student's *t* test, $t = 4.097$, df = 3). Left panel: representative images showing platelet adhesion. In each graph, each symbol represents 1 individual. In all images, scale bars = 100 μm. Independent experiments correspond to different mice. Source data are provided as a Source Data file.

related proteins, PIEZO-1 or tubulin family members was unchanged in *Mical1*⁻/⁻ platelets (Supplementary Fig. 5d, e), consistent with the idea that MICAL1 may directly control actin levels through oxidation.

To test whether the increased levels of F-actin observed in *Mical1*⁻/⁻ platelets underlie their adhesion and stability defects, we treated whole blood with 125 nM of LatA (note that the concentration of LatA is reduced, compared to the concentrations used in Fig. 1). At these relatively low doses of LatA, no effect on adhesion or stability of *Mical1*⁺/⁺ platelets was observed (Fig. 5f, g). In contrast, this LatA treatment restored normal adhesion and stability of *Mical1*⁻/⁻ platelets to levels comparable to controls (Fig. 5f, g). Thus, MICAL1 deficiency induces excessive F-actin polymerization (see also below), which is responsible for the defects in adhesion and stability of *Mical1*⁻/⁻ platelets.

Altogether, our data reveal that MICAL1-mediated F-actin disassembly promotes platelet adhesion and stability on VWF in the presence of arteriolar shear.

## MICAL1 promotes stable platelet adhesion to VWF by increasing resistance to shear-induced morphological changes

The establishment of stable adhesion contacts between platelets and VWF requires mechanisms that can withstand the mechanical forces generated by blood flow, in particular the mobilization of the open canalicular system (OCS), which extends membrane tethers to mediate firm adhesion and spreading[5,6,56]. We hypothesized that the defects in *Mical1*⁻/⁻ platelet adhesion and stability, which are associated with abnormally high levels of F-actin as described above, could be the consequence of defects in the morphological changes that occur under high shear. To test this hypothesis, we measured the kinetics of

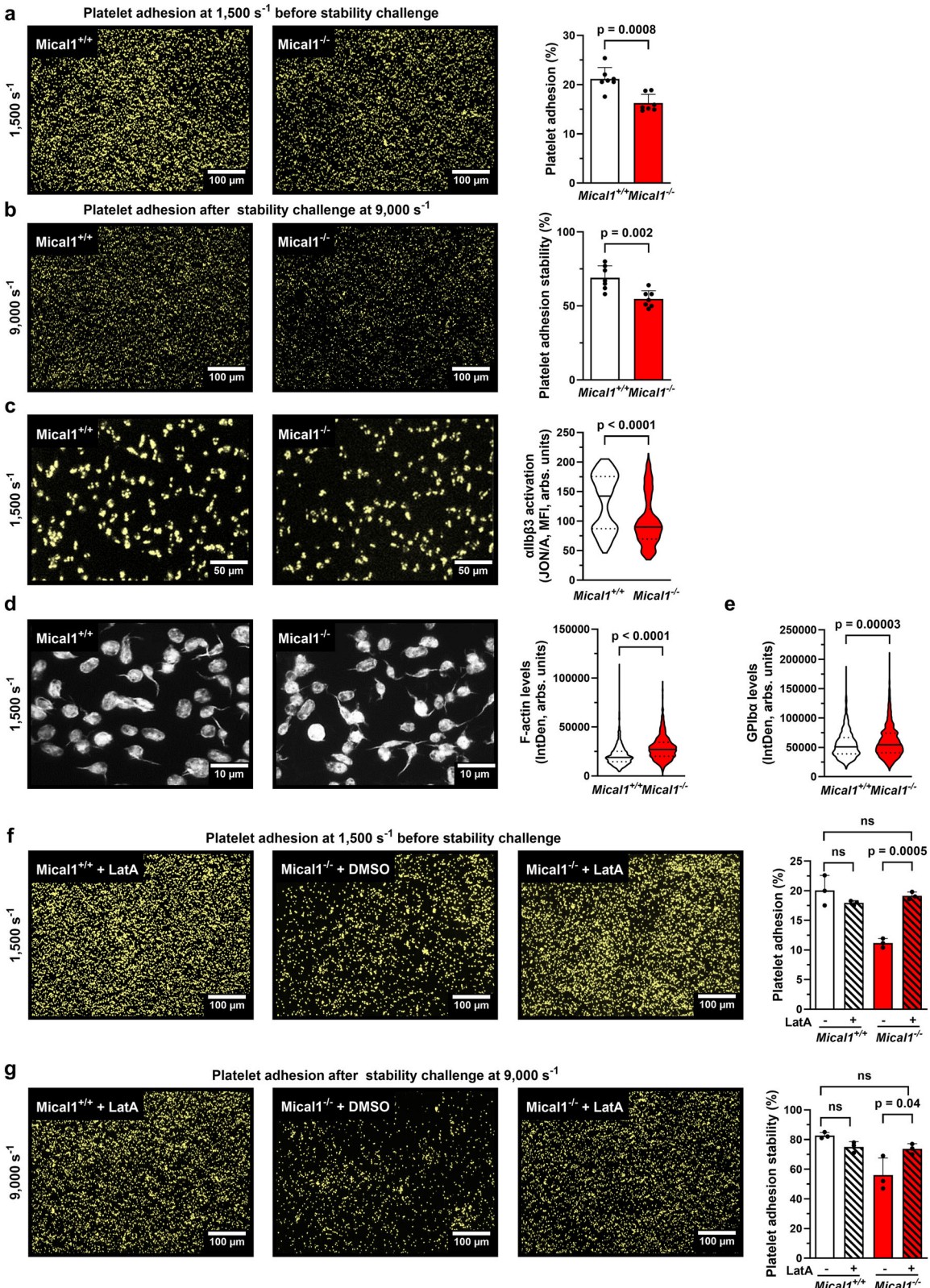

tether dynamics in real time at a shear rate of 1500 s⁻¹ (Supplementary Fig. 1a, Supplementary Movies 4 and 5). As previously described[5], tethers formed in control platelets and underwent a retraction phase (Fig. 6a–c, black). In contrast, one third of $Mical1^{-/-}$ platelets exhibited elongated tethers that failed to retract (Fig. 6a–c, red). Overall, tethers retracted by $84 \pm 4\%$ in $Mical1^{+/+}$ platelets, whereas no significant retraction was observed in $Mical1^{-/-}$ platelets (by $4 \pm 18\%$) during the

first 3 min. However, after 6 min, no difference in individual tether length was measured in $Mical1^{-/-}$ platelets ($Mical1^{-/-}$: $1.62 \pm 0.11$ μm vs. $Mical1^{+/+}$: $1.82 \pm 0.07$ μm), indicating that the retraction phase was ultimately achieved in the absence of MICAL1. Nonetheless, $Mical1^{-/-}$ platelets exhibited larger surface compared to $Mical1^{+/+}$ platelets (Fig. 6d) and this was associated with an increase in F-actin levels within $Mical1^{-/-}$ tethers compared to $Mical1^{+/+}$ tethers (Fig. 6e). To

**Fig. 5 | Control of F-actin disassembly by MICAL1 supports shear-dependent platelet adhesion. a**, **b**, **f**–**g** *Mical1*[+/+] or *Mical1*[−/−] whole blood was perfused on recombinant mouse VWF matrix. **a** Platelet adhesion (% of the surface coverage) (mean ± SD, N = 7 independent experiments from different mice, two-tailed unpaired Student's *t* test, *t* = 4.420, df = 12), and **b** platelet adhesion stability (mean ± SD, N = 7 independent experiments, two-tailed unpaired Student's *t* test, *t* = 3,813, df = 12). Left panels: representative images before (**a**) and after (**b**) stability challenge. Scale bars = 100 μm. **c** Integrin αIIbβ3 activation of *Mical1*[+/+] or *Mical1*[−/−] platelets under flow (1500 s[−1]) (mean ± SD, N = 600 platelet, from 4 independent experiments corresponding to different mice, two-tailed unpaired Student's *t* test, *t* = 12.30, df = 1198). Left panels: representative images showing JON/A staining. Scale bars = 10 μm. **d** F-actin levels in platelets (integrated density (IntDen) ± SD, N = 819 platelets from 3 independent experiments from different mice, two-tailed unpaired Student's *t* test, *t* = 3.110, df = 4) (Left panels: representative images of F-actin. Scale bars = 5 μm). **e** GPIbα expression (integrated density (IntDen)) ± SD, N: *Mical1*[+/+] = 1034; *Mical1*[−/−] = 1011 platelets from 3 independent experiments from different mice, two-tailed unpaired Student's *t* test, *t* = 0.4695, df = 4). **f**, **g** Rescue experiment of platelet adhesion in the presence of LatA. *Mical1*[+/+] or *Mical1*[−/−] blood treated with DMSO or LatA (125 nM) was perfused on recombinant mouse VWF matrix. **f** Platelet adhesion (% of the surface coverage) (mean ± SD, N: *Mical1*[+/+]/DMSO = 4, *Mical1*[+/+]/LatA = 4, *Mical1*[−/−]/DMSO = 3, *Mical1*[−/−]/LatA = 3 independent experiments from different mice, one-way ANOVA with Tukey post-hoc test, F = 2.007, df = 8) and **g** stable platelet adhesions were measured (mean ± SD, N: *Mical1*[+/+]/DMSO = 4, *Mical1*[+/+]/LatA = 4, *Mical1*[−/−]/DMSO = 3, *Mical1*[−/−]/LatA = 3 independent experiments from different mice, one-way ANOVA with Tukey post-hoc test, F = 0.9851, df = 8). Left panel: representative images showing platelet adhesion before (**f**) and after (**g**) stability challenge. Scale bars = 100 μm. ns not significant. Source data are provided as a Source Data file.

further assess the behavior of the tethers, we next challenged their stability by increasing the shear rate to 9000 s[−1]. Interestingly, we observed a second wave of tether elongation in platelets from both mouse genotypes, but we measured a significantly greater length of tethers in *Mical1*[−/−] (4.02 ± 0.34 μm) compared to *Mical1*[+/+] platelets (2.35 ± 0.15 μm) (Fig. 6f). Thus, the absence of MICAL1 both slows down tether retraction and leads to a failure to properly limit tether elongation induced by high shear.

Scanning electron microscopy confirmed that at a shear rate of 9000 s[−1], *Mical1*[−/−] platelets displayed a more elongated morphology (i.e. decreased circularity index), unveiling a susceptibility to deformation that is not restricted to the tethers (Fig. 6g). We further confirmed this observation in the human megakaryocytic cell line DAMI expressing GPIbα. MICAL1-depleted cells showed a marked increase in size, shear strain and a loss of circularity with increasing shear compared to control cells (Supplementary Fig. 6). Interestingly, we observed "ghost" platelets with membrane-free cytoskeletal remnants attached to the matrix specifically for *Mical1*[−/−] platelets, indicative of bursting and likely the consequence of a decrease in resistance to deformation upon MICAL1 deficiency (Fig. 6h). In contrast to what we observed with high shear, no differences in platelet spreading, filopodia formation, or lamellipodia extension on VWF or fibrinogen matrices were found in the absence of hemodynamic condition in *Mical1*[−/−] platelets compared to control platelets (Supplementary Fig. 7a–d). Additionally, ultrastructural analysis revealed no differences in OCS structure between *Mical1*[−/−] and *Mical1*[+/+] platelets (Supplementary Fig. 7e), indicating that defects in tethers are not due to impaired membrane biogenesis. Thus, MICAL1 deficiency impairs morphological changes only when platelets are subjected to shear on VWF.

Altogether, we conclude that MICAL1 helps to counteract shear-induced morphological changes, which is necessary for efficient platelet adhesion and stability on VWF.

## MICAL1 promotes F-actin disassembly in the GPIb-IX-V complex in a mechano-sensitive manner, enabling GPIbα translocation into lipid rafts and increased association with VWF

Since F-actin accumulation impairs platelet adhesion, stability, and resistance to deformation under shear in *Mical1*[−/−] platelets, we investigated potential defects in VWF-GPIbα interaction under static and shear conditions in these cells. We found that VWF binding to platelets without shear (i.e. in suspension) was comparable between *Mical1*[−/−] and *Mical1*[+/+] platelets (Fig. 7a, b). In addition, co-immunoprecipitation of GPIbα in resting and VWF/botrocetin-activated platelets revealed similar β-actin association for both genotypes (Fig. 7c, d). These results indicate that MICAL1 is dispensable for VWF-GPIbα interaction and F-actin association with the GPIb-IX-V complex in the absence of shear. We next investigated the role of MICAL1 specifically in VWF-GPIbα interaction and under shear by assessing platelet rolling and measuring

translocation velocities on immobilized VWF at a shear rate of 1500 s[−1] after blockade of αIIbβ3 integrins. The number of rolling platelets from *Mical1*[−/−] mice was significantly decreased by 50%, compared to control mice (Fig. 7e). Furthermore, *Mical1*[−/−] platelets exhibited significantly higher translocation velocities (2.06 ± 1.04 μm/s) than *Mical1*[+/+] platelets (1.41 ± 0.89 μm/s), indicating reduced VWF-GPIbα interactions (Fig. 7f, Supplementary Movies 6 and 7). Thus, as observed upon actin stabilization by Jasp (Fig. 1), VWF-GPIbα interactions were impaired upon MICAL1 depletion. These results demonstrate that MICAL1 promotes VWF-GPIbα interactions under high shear.

Next, we determined whether MICAL1 deficiency altered the amount of F-actin associated with the GPIb-IX-V complex under shear. GPIbα co-immunoprecipitation after blood perfusion at a shear rate of 1500 s[−1] revealed a > 3-fold increase in β-actin associated with GPIbα in *Mical1*[−/−] platelets, compared to control platelets (Fig. 7g). To test whether F-actin accumulation in the GPIb-IX-V complex underlies defects in VWF-GPIbα interactions, we treated *Mical1*[−/−] platelets with a low dose of LatA (125 nM). Remarkably, this treatment both reduced F-actin associated to GPIbα under shear (Fig. 7h) and restored the rolling velocities of *Mical1*[−/−] platelets to wild-type levels, whereas LatA had no effect on control platelets (Fig. 7i, Supplementary Movies 8–11). These results show that the abnormal accumulation of F-actin in the GPIb-IX-V complex in *Mical1*[−/−] platelets is responsible for the reduced VWF-GPIbα interactions, explaining the platelet adhesion defects observed upon high shear.

Finally, we explored how, at the mechanistic level, F-actin accumulation in the GPIb-IX-V complex could reduce the interaction between VWF and the GPIb-IX-V complex in *Mical1*[−/−] platelets. It is known that the exposure of platelets to high shear leads to the translocation of GPIbα to lipid rafts, thereby promoting GPIb-IX-V complex clustering, efficient interaction with VWF and platelet adhesion[12–15]. Consistently, cholesterol depletion induced by methyl-β-cyclodextrin treatment completely disrupted platelet adhesion and stability on VWF (Supplementary Fig. 8), as previously described[12,15]. It is now well established that F-actin plays a critical role in organizing these lipid domains and controls signaling from receptors in these domains[57–59]. We thus hypothesized that the excess of F-actin associated with GPIbα in MICAL1-deficient platelets might modify GPIbα association into lipid rafts enriched in cholesterol. Using cholera toxin (CTxB) as a marker for lipid rafts[60], we found a decreased localization of GPIbα in lipid rafts in *Mical1*[+/+] platelets treated with Jasp (Fig. 8a) and in *Mical1*[−/−] platelets, compared to *Mical1*[+/+] platelets (Fig. 8b). This reduction in colocalization occurred in both the platelet body and the tethers (Supplementary Fig. 9). Importantly, rescue experiments using low concentrations of LatA fully restored the presence of the GPIb-IX-V complex in lipid rafts in *Mical1*[−/−] platelets (Fig. 8c), demonstrating that the accumulation of F-actin within the GPIb-IX-V complex upon MICAL1 deficiency is responsible for the defects in the GPIb-IX-V complex association in lipid rafts.

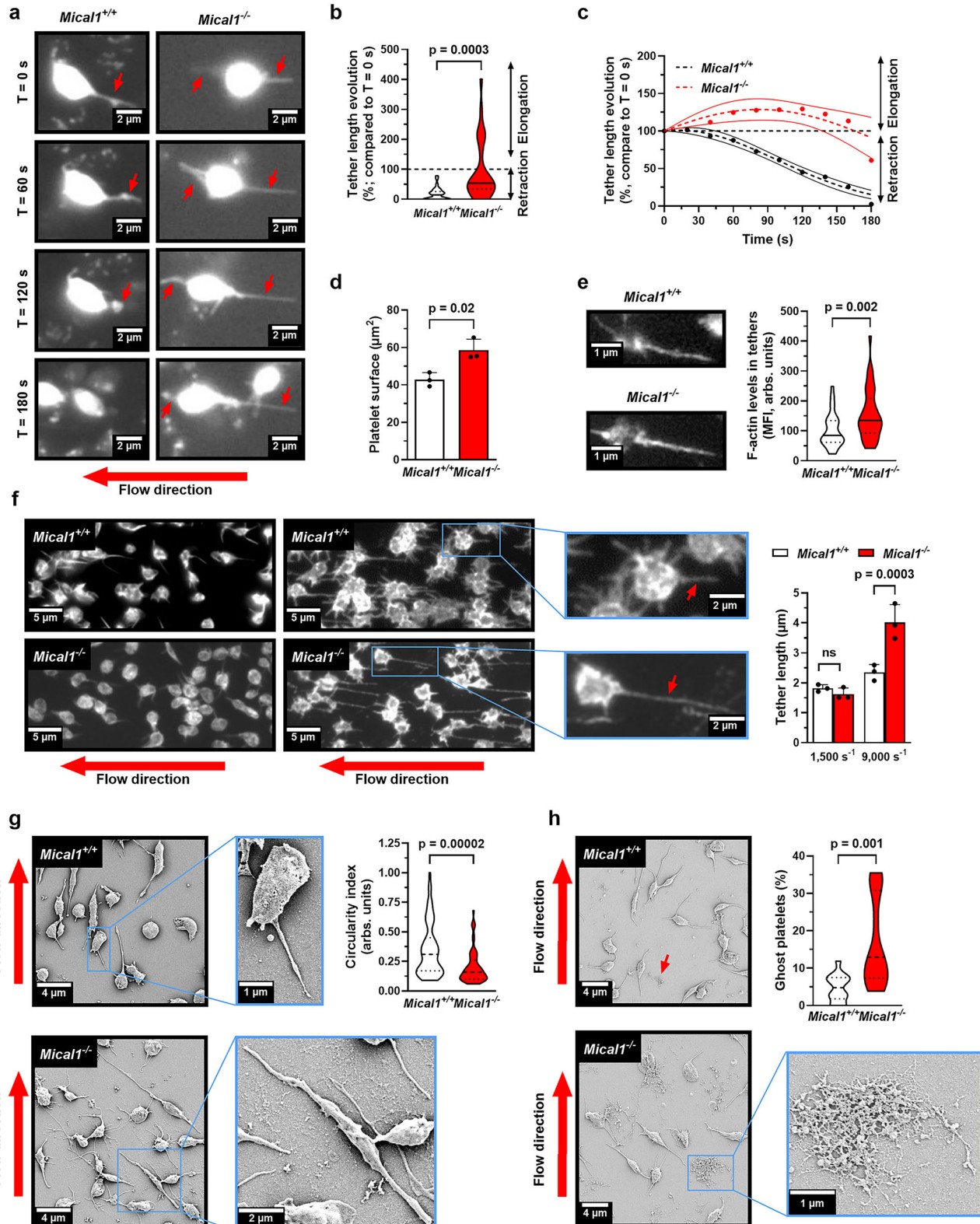

We conclude that MICAL1 plays a key role in disassembling F-actin associated with GPIbα, allowing its translocation to lipid rafts, which in turn promotes efficient platelet interaction with VWF under high shear.

## Discussion

Here, we report that F-actin disassembly by the enzyme MICAL1 at the GPIb-IX-V complex is critical for promoting stable platelet adhesion to VWF under hemodynamics conditions and thus for thrombus stability in vivo. In the absence of MICAL1, F-actin accumulates abnormally within the GPIb-IX-V complex, VWF-GPIbα interactions are impaired and platelet resistance to shear-induced morphological changes is reduced. As a result, platelet adhesion and arterial thrombus stability are impaired. Mechanistically, the absence of MICAL1 reduces the translocation of GPIbα to lipid rafts due to excessive F-actin levels. Thus, this work reveals that limiting F-actin levels after platelet

**Fig. 6 | MICAL1 promotes stable platelet adhesion to VWF by increasing resistance to shear-induced morphological changes. a, b** Whole blood from *Mical1*⁺/⁺ or *Mical1*⁻/⁻ mice was perfused over a recombinant mouse VWF-coated surface at 1500 s⁻¹. **a** Representative time-lapse images showing individual platelet tether formation. Red arrows indicate tethers. Scale bars, 2 μm. See Supplementary Movies 4 and 5. **b** Tether elongation was quantified as [(length at 3 min/length at 0 s) ×100]; mean ± SD, $N = 30$ tethers from 3 independent experiments (Welch's $t$-test, $t = 4.105$, df = 31.52). **c** Tether length was tracked over time; $N = 30$ tethers from 3 experiments; mean ± 95% CI (two-way ANOVA with Šídák's test, F(9, 504) = 4.375; $p = 0.003$ at 80 s; $p < 0.001$ from 100 to 160 s). **d–f** Unstained *Mical1*⁺/⁺ or *Mical1*⁻/⁻ blood was perfused on VWF. Post-perfusion, coverslips were fixed and stained with phalloidin and anti-integrin β3. **d** Platelet surface area measured in 3 experiments (273 platelets/experiment); mean ± SD (Student's $t$ test, $t = 3.938$, df = 4). **e** F-actin quantification in tethers at 1500 s⁻¹ using phalloidin (*Mical1*⁺/⁺ = 39, *Mical1*⁻/⁻ = 37

tethers; Welch's $t$ test, $t = 3.261$, df = 61.62). Left: representative images with actin-positive tethers (red arrows). Scale bars, 1 μm. **f** Tether length measured from at least 254 platelets/experiment ($N = 3$); mean ± SD (two-way ANOVA with Fisher's LSD test, F(1,8) = 22.09). Left: representative β3-labeled tethers at 1500 and 9000 s⁻¹. Scale bars, 5 μm. **g** Scanning electron microscopy (SEM) images of platelets from 9000 s⁻¹ flow assays. Circularity index quantified in 2 experiments ($n = 2$ mice/condition pooled); *Mical1*⁺/⁺ = 65, *Mical1*⁻/⁻ = 63 platelets (Student's $t$ test, $t = 4.495$, df = 126). Scale bars: 4 μm (overview), 1–2 μm (zoom). **h** Frequency of "ghost" platelets per field assessed by SEM in 2 experiments (*Mical1*⁺/⁺ = 743, *Mical1*⁻/⁻ = 609 platelets; Student's $t$ test, $t = 4.495$, df = 126). Representative SEM images with zooms shown. Scale bars: 4 μm and 2 μm. Violin plots display medians (central line) and interquartile ranges (25th–75th percentiles). ns: not significant. Source data are provided as a Source Data file.

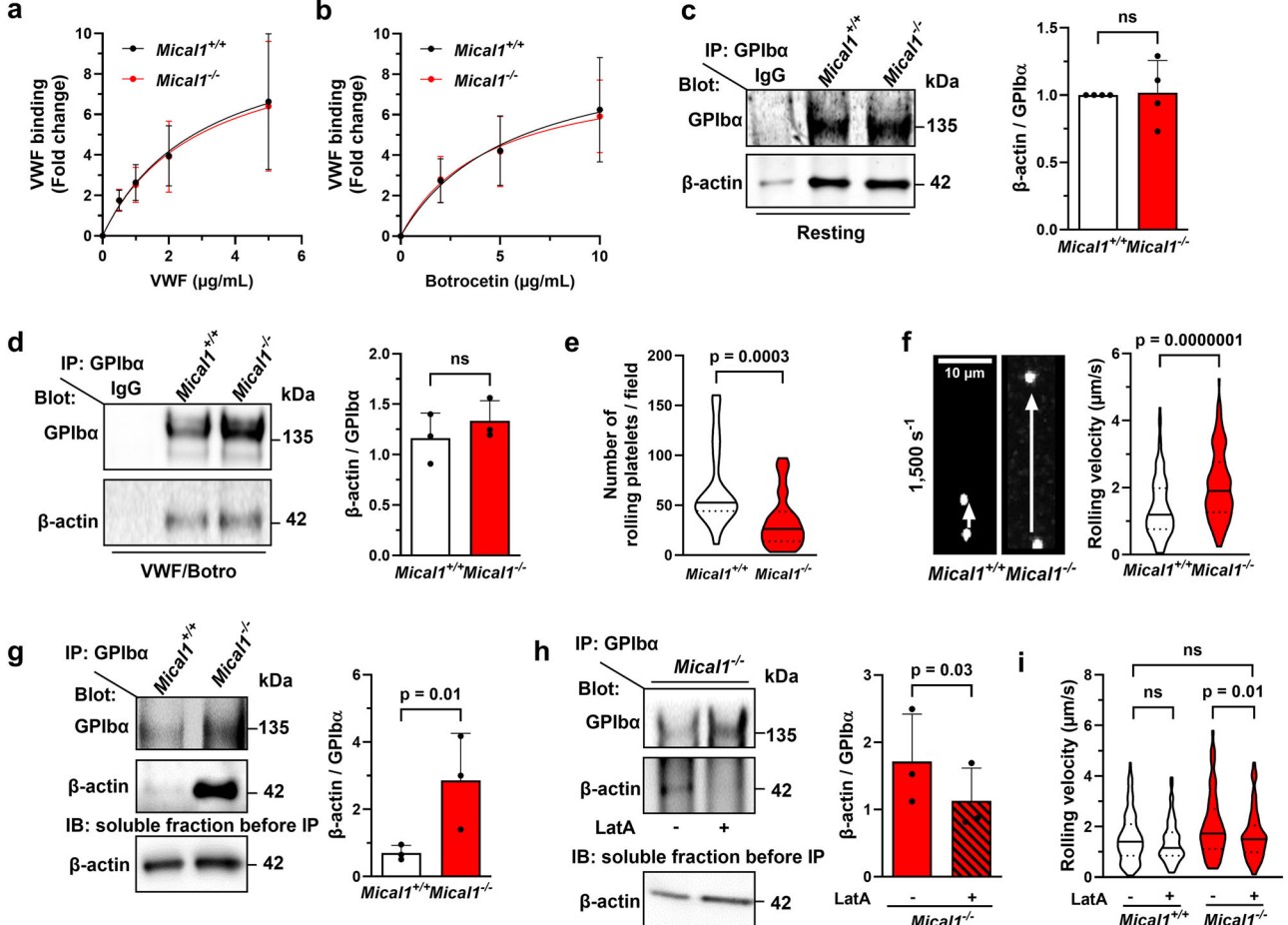

**Fig. 7 | MICAL1 promotes F-actin disassembly in the GPIb-IX-V complex in a mechano-sensitive manner, increasing GPIbα association with VWF. a, b** VWF binding assessed by flow cytometry in *Mical1*⁺/⁺ and *Mical1*⁻/⁻ platelets. **a** Platelets were activated with 5 μg/mL botrocetin and increasing concentrations of recombinant mouse VWF. **b** Platelets were activated with 0, 5 or 10 μg/mL botrocetin in the presence of 10 μg/mL recombinant VWF. Data are mean ± SD from $N = 3$ (**a**) or 4 (**b**) independent experiments (two-way ANOVA with Šídák's test; (**a**) F(4, 20) = 0.004839; (**b**) F(3,24) = 0.02602). **c, d** GPIbα immunoprecipitation from lysates of washed platelets. **c** Resting platelets; **d** platelets stimulated in static conditions with recombinant VWF (10 μg/mL) and botrocetin (5 μg/mL). Immunoblots show GPIbα and β-actin (mean ± SD, $N = 4$; Student's $t$ tests: **c** $t = 0.1464$, df = 6; **d** $t = 0.9347$, df = 4). **e, f)** Platelet rolling on VWF under shear (1500 s⁻¹). **e** Number of rolling platelets per field (*Mical1*⁺/⁺ = 30, *Mical1*⁻/⁻ = 30 fields; $t = 3.864$, df = 58).

**f** Rolling velocity (*Mical1*⁺/⁺ = 131, *Mical1*⁻/⁻ = 129 platelets; $t = 5.452$, df = 258). Left: representative time-lapse images (10 s). Scale bar, 10 μm. See Supplementary Movies 6 and 7. **g** GPIbα immunoprecipitation from platelet lysates after flow. Soluble fractions were immunoblotted for β-actin as control. Mean ± SD from $N = 3$ experiments (ratio paired Student's $t$ test, $t = 8.084$, df = 2). **h** GPIbα immunoprecipitation after flow in *Mical1*⁻/⁻ platelets treated with LatA (125 nM) or DMSO. Soluble fractions were immunoblotted for β-actin as control. Mean ± SD from $N = 3$ experiments (ratio paired Student's $t$ test, t = 5.888, df = 2). **i** Platelet rolling rescue experiment with LatA. *Mical1*⁺/⁺ and *Mical1*⁻/⁻ platelets were treated with LatA (125 nM) or DMSO and perfused under flow. Velocity measured from $N = 3$ experiments (*Mical1*⁺/⁺ + DMSO = 183, +LatA = 141; *Mical1*⁻/⁻ + DMSO = 134, +LatA = 138 platelets; one-way ANOVA with Tukey's test, F = 10.05, df = 581). See Supplementary Movies 8–11. ns: not significant. Source data are provided as a Source Data file.

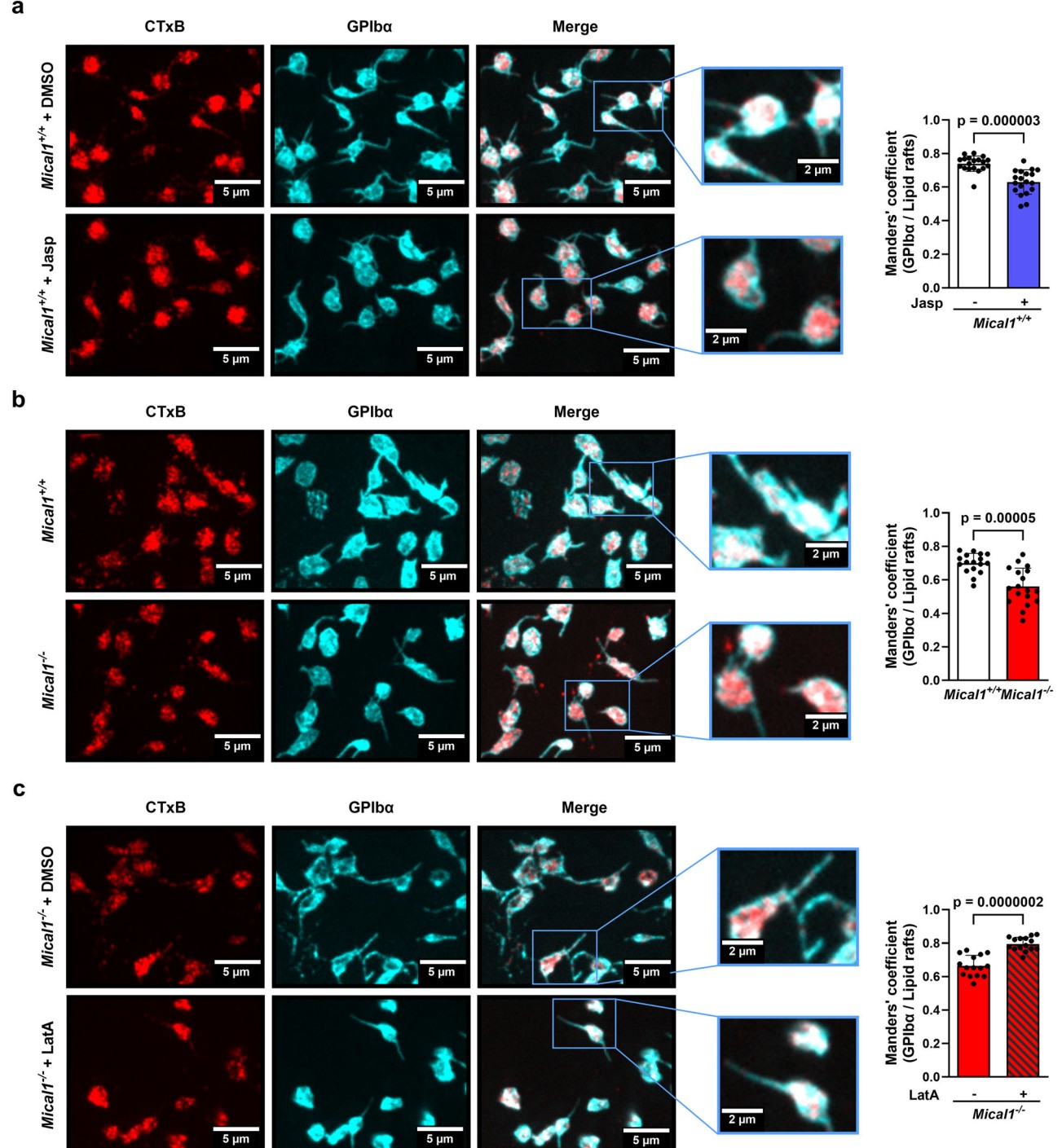

**Fig. 8 | MICAL1 induced F-actin disassembly in the GPIb-IX-V complex enable GPIbα translocation into lipid rafts. a–c** Colocalization of lipid rafts (CTxB, Red) and GPIbα (Green) in *Mical1*⁺/⁺ and *Mical1*⁻/⁻ platelets after flow assays at 1500 s⁻¹ from 3 independent experiments corresponding to different mice. Left panels: representative images (scale bars: 5 µm) and zoom (scale bars: 2 µm). **a** Manders' overlap coefficient for *Mical1*⁺/⁺ platelets treated with either Jasp (2 µM) or DMSO as control (mean ± SD, *N* = 19 fields from 3 independent experiments for both

conditions, two-tailed unpaired Student's *t* test, *t* = 5.563, df = 36). **b** Manders' overlap coefficient for *Mical1*⁺/⁺ vs *Mical1*⁻/⁻ platelets (mean ± SD, *N* = 17 fields from 3 independent experiments for both genotypes, two-tailed unpaired Student's *t* test, *t* = 4.642, df = 32). **c** Manders' overlap coefficient for *Mical1*⁻/⁻ platelets treated with either a rescue dose of LatA (125 nM) or DMSO as control (mean ± SD, *N* = 15 fields from 3 independent experiments for both genotypes, two-tailed unpaired Student's *t* test, *t* = 6.792, df = 28).

activation is a key step for mechano-dependent GPIbα signaling under shear to efficiently stabilize platelets at the site of injury (Fig. 9).

The importance of F-actin dynamics in platelet function after activation by VWF under shear is largely unknown. To date, cortical actin has primarily been implicated in structural and platelet shape

changes[61]. Furthermore, initial VWF-GPIbα interactions result in a global assembly of actin filaments in platelets[16]. Here, we show that F-actin accumulates in the GPIb-IX-V complex upon VWF-GPIbα binding (Fig. 2). This is presumably important for controlling the lateral mobility of the GPIb-IX-V complex in the plane of the plasma

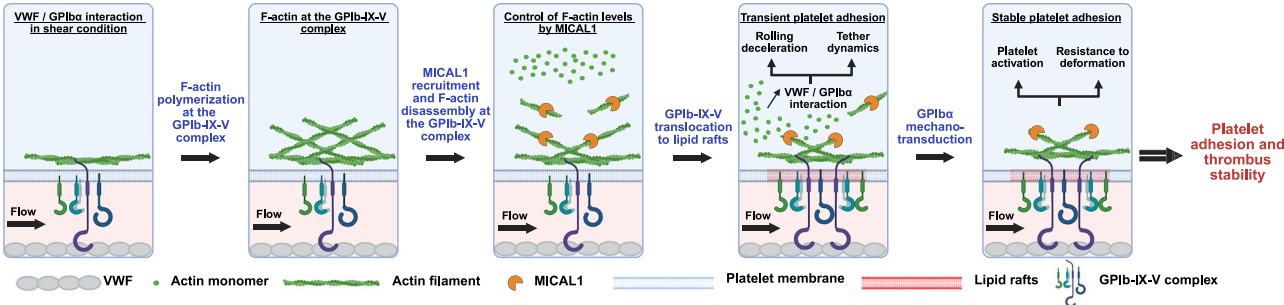

**Fig. 9 | F-actin disassembly by the oxidoreductase MICAL1 promotes mechano-dependent VWF-GPIbα interaction in platelets.** GPIbα interaction with VWF under high shear stress induces F-actin and MICAL1 recruitment in the GPIb-IX-V complex, which limits F-actin associated within the GPIb-IX-V complex. This process both facilitates receptor translocation to lipid rafts, which strengthens its interaction with VWF and promotes the retraction of platelet membrane tethers, resulting in increased platelet adhesion. Ultimately, GPIbα mechanotransduction activates platelets, increasing their resistance to deformation and enabling stable platelet adhesion and thrombus stability. *Created in BioRender. Solarz, J. (2025)* https://BioRender.com/hfvgvsz.

membrane in a Filamin A-dependent manner in order to increase GPIbα and VWF interactions[13]. In addition, internal forces acting through F-actin are likely transmitted by the GPIb-IX-V complex to strengthen interactions with VWF[19]. Using pharmacological approaches, we showed that an appropriate level of F-actin is essential for proper platelet interaction with VWF, specifically under high shear and F-actin levels must be tightly regulated for optimal stable platelet adhesion (Fig. 1) notably to control interactions between cytoskeleton-associated proteins and receptors[62]. First, reducing F-actin levels with LatA promotes platelet adhesion and decreases velocities of rolling platelets, consistent with previous findings that depolymerizing F-actin leads to the formation of membrane tethers[5,6]. Conversely, actin stabilization with Jasp impairs platelet rolling and adhesion under hemodynamic conditions and greatly reduces stable platelet attachment to VWF. This is consistent with a proposed role for F-actin in limiting VWF-GPIbα signaling[63]. Altogether, this shows that F-actin levels must be tightly regulated for optimal stable platelet adhesion. We identified the actin disassembling enzyme MICAL1 as one of the key mechanisms that prevents F-actin levels from exceeding a threshold that is detrimental to platelet function. Functionally, knocking-out MICAL1 in megakaryocytes and platelets resulted in prolonged bleeding time, impaired thrombus stability and embolization in vivo. We observed that the embolization phenotype became more pronounced with increasing shear rates. Specifically, no phenotype was detected in mesenteric venules, which are characterized by low shear rates, while a significant increase in occlusion was observed in mesenteric arterioles, where shear rates are higher. This shear-dependent phenotype suggests a preferential involvement of the VWF/GPIb axis, known to be activated under high shear conditions, rather than the collagen/GPVI axis, which typically operates under low to moderate shear conditions. The absence of a phenotype at low shear further supports the idea that MICAL1 functions within a shear-sensitive pathway that relies on VWF-GPIb interactions. These results suggest that MICAL1 could represent a promising therapeutic target for pathologies characterized by VWF hyperactivation, such as in patients who have undergone transcatheter aortic valve implantation (TAVI) or those with type 2B von Willebrand disease. Furthermore, from a clinical perspective, the existence of a *MICAL1* variant associated with bleeding suggests that the F-actin limitation by MICAL1 plays a role in human platelet function[64].

At the cellular level, our functional assays revealed three functions of MICAL1 in platelets through its role in actin disassembly. Firstly, MICAL1 plays a negative role in platelet secretion and aggregation (Fig. 4 and Supplementary Fig. 3), consistent with a role of F-actin and its dynamics in promoting secretion and aggregation[65–69]. Of note, this cannot explain the defects observed in vivo in *Mical1*−/− mice, which

emphasize a dominant role of MICAL1 in shear-dependent adhesive and thrombotic functions.

Secondly, MICAL1 limits the extension of membrane tethers that form under shear stress and promotes their retraction, thereby promoting platelet adhesion, as observed in vivo (Fig. 6). Remarkably, the tethers in *Mical1*−/− platelets are longer and fail to retract properly at an arteriolar shear rate of 1500 s⁻¹, likely due to the observed abnormally high levels of F-actin within the tethers. Regulation of F-actin levels by MICAL1 also appears to be critical for platelet resistance to higher shear rates and associated membrane resilience (Fig. 6). Similarly, platelets deficient in the actin-binding protein Coactosin-like 1 - which prevents F-actin depolymerization - displayed altered morphology with reduced adhesion and thrombus formation on VWF but how this is achieved mechanistically has not been described[70]. Altogether, our results highlight the central role of the actin cytoskeleton in cell resistance and deformation under hemodynamic conditions, which contributes to stable platelet adhesion.

Thirdly, MICAL1 plays a critical role in promoting VWF-GPIbα interaction and signaling, which also explains MICAL1's function in platelet adhesion. The GPIb-IX-V complex is responsible for the initial interaction between platelets and VWF, which helps to slow down platelet and thus promote more robust binding to the lesion site through integrin activation[71]. Specifically, we show that an appropriate level of F-actin, regulated by MICAL1, significantly contributes to the affinity and binding strength of platelets to VWF induced by high shear (Figs. 5 and 7). The importance of actin disassembly is underscored by the observation that MICAL1-deficient platelets exhibit increased platelet rolling velocity - reflecting reduced VWF-GPIbα interaction under blood flow - which was rescued by reducing F-actin levels by LatA (Fig. 7). Importantly, MICAL1-mediated actin disassembly strengthens VWF-GPIbα interactions specifically under high shear (Fig. 5 and Supplementary Fig. 4), indicating that MICAL1 is important for mechano-dependent signaling. As a first step, MICAL1 is rapidly recruited to the GPIb-IX-V complex by F-actin upon platelet activation (Fig. 2). Then, MICAL1 limits the association of F-actin with GPIbα, likely thanks to its well-established actin disassembly activity involving actin oxidation, which promotes GPIbα interaction with VWF (Fig. 7). Future investigations will be needed to understand how MICAL1 is activated under shear. This is likely to be achieved downstream of the initial unfolding of GPIbα by shear by releasing MICAL1 from its autoinhibited conformation through protein binding to its C-terminal domain[30,32,72–76]. Mechanistically, our data show that the disassembly of F-actin by MICAL1 promotes the translocation of GPIbα to lipid rafts (Fig. 8), which explains why the absence of MICAL1 reduces VWF-GPIbα interaction under high shear but does not affect VWF binding to platelets

under static conditions (Fig. 7). Indeed, depletion of membrane cholesterol, which disrupts lipid rafts, significantly affected platelet adhesion to VWF under high shear, without altering VWF binding under static conditions[6]. At the molecular level, the excess of F-actin observed in MICAL1-deficient platelets could disrupt the lateral mobility of the GPIb-IX-V complex, hindering its translocation to lipid rafts, thereby reducing its interaction with VWF and reducing αIIbβ3 integrin activation (Fig. 5) necessary to stabilize the adhesion. This hypothesis is based on the evidence that the clustering of and the organization of lipid rafts into more active signaling platforms are influenced by interactions with F-actin and dynamic rearrangements of the cytoskeleton[57–59] including upon platelet activation[77], and these domains play a crucial functional role for the signaling of numerous receptors[78,79]. Altogether, MICAL1-mediated F-actin disassembly is important for regulating GPIbα association with lipid rafts and its interaction with VWF under shear. Of note, the changes in F-actin disassembly during platelet activation are reversible[80] and likely depend on a dynamic balance between MICAL1-mediated oxidation and Methionine Reductase (MSRB)-mediated reduction of actin, as observed during bristle development in drosophila or cytokinesis in human cells[24,30,37,81]. In a broader perspective, this highlights a critical role of actin cytoskeletal remodeling and lipid raft organization in mechanotransduction.

In conclusion, our work shows that an optimal level of actin polymerization is crucial for platelet adhesion and stability at the injured blood vessels. This is achieved through the recruitment of MICAL1 to the GPIb-IX-V complex to promote GPIbα binding to VWF in a mechano-dependent manner, revealing a role for regulated actin oxidation in platelet biology.

## Methods
A list of reagents and antibodies can be found in Supplementary Data 1.

### Animals
This project was approved by the local ethical committee CEEA26 and the French government under the number APAFIS#25086-2020032312267714. *Mical1*^fl/fl mice were kindly provided by R. Jeroen Pasterkamp (Utrecht, Netherlands)[34], and were crossed with mice carrying the Cre recombinase under the control of the platelet factor 4 (*Pf4*) promoter[82], generating conditional wild-type (WT, *Mical1*^fl/fl hereon referred to as *Mical1*^+/+) and knock-out (KO, *Mical1*^fl/fl;Pf4-Cre hereon referred to as *Mical1*^−/−) MICAL1 mice in megakaryocytes and platelets. Genotyping of mice was performed by PCR with 5′GTTGAAAGATGTGGACCAGGA3′ and 5′GGACAGGGAGAGCAGA-GACTT3′ primers for floxed *Mical1* and with 5′CCCATACAGCA-CACCTTTTG3′ and 5′TGCACAGTCAGCAGGTT3′ primers for *Pf4*-Cre with the KAPA2G kit. For mice studies, housing and experiments were done as recommended by French regulations and the experimental guidelines of the European Community. Animals were provided with food and water ad libitum; bedding was enriched with wood pieces for gnawing and kraft paper for nesting. The temperature and humidity in the housing rooms were strictly controlled and maintained within the following ranges: 21 ± 3 °C (18–24 °C) for temperature, and 50–60% ± 10% (45–65%) for relative humidity. A 12 h/12 h light/dark cycle was maintained throughout the study. All procedures were performed with constant attention to minimizing discomfort and pain (monitoring of endpoints, use of anesthesia). Animals were observed daily to ensure their well-being. Furthermore, all personnel involved in the project were technically qualified and received continuous training in animal experimentation practices. To minimize stress, mice were removed from the housing area only immediately before experimentation. Animals were never housed alone (a maximum of five per cage, with aggressive animals being excluded). Anesthesia, experimental procedures, and euthanasia were carried out in rooms completely separate from the housing area. Both male and female mice

were analyzed (4-16-week-old). In all experiments, mice were anesthetized either with a mixture of ketamine and xylazine (100 mg/kg and 10 mg/kg, respectively) injected intraperitoneally or with inhaled isoflurane gas.

### Blood parameters
For assessment of platelet count, size and basic blood parameters, mouse blood was collected in tubes containing EDTA (10 mM) using heparinized capillaries, prior to measurement of the various hematological parameters using an automated cell counter (Scil Vet ABC Plus, Horiba Medical).

### Measure of bleeding time (tail clip)
Bleeding time assays were performed on 8-week-old mice by cutting off the tip of the tail (3 mm from the tip, tail clip assay) and immediately immersing it in saline[83]. Bleeding time was measured from the moment of transection until the arrest of bleeding. Observation was stopped at 600 s when bleeding did not cease.

### FeCl₃-induced thrombosis model
For the carotid model, 10- to 16-week-old mice were used. After a lateral deposition of $FeCl_3$ solution (10%) to the right carotid for 2 min, the vessel was then washed 3 times with NaCl 0.9%. Thrombus formation was monitored in real-time by measuring the blood flow with a perivascular flow probe (TS420 Perivascular Flow Module, Transonic System Inc, Ithaca, NY). The equipment was calibrated using a standard flow meter in mL/min. The waveform of the blood flow was recorded using LabChart Reader 8 software (AD instrument).

For the mesenteric model, $FeCl_3$ injury was induced in 4- to 5-week-old mice. To facilitate visualization of thrombus formation, platelets were fluorescently labeled in vivo by intravenous injection of rhodamine 6G (3.3 mg/kg). After topical deposition on the mesenteric vessels of $FeCl_3$ solution (10%), thrombus growth was monitored in real-time with an inverted epifluorescence microscope (×10; Nikon Eclipse TE2000-U). If no occlusion was observed after 60 min, the occlusion time was recorded as over 60 min.

### Flow chamber assay models
For all models, whole mouse blood was collected with hirudin (400 U/mL).

Thrombus formation and stability was evaluated on collagen matrix (50 μg/mL) with platelet labeled with rhodamine 6 G (10 μg/mL). Blood was perfused at a shear rate of 1500 s⁻¹ for 3 min, then washed with Tyrode's buffer (137 mM NaCl, 2 mM KCl, 0.3 mM NaH₂PO₄, 1 mM MgCl₂, 5.5 mM glucose, 5 mM N-2-hydroxyethylpiperazine-N′−2-ethanesulfonic acid, 12 mM NaHCO₃, pH 7.3) for 2 min at a shear rate of 1500 s⁻¹. Thrombus stability was assessed on stable thrombi by submitting them to stability challenge protocol consisting of perfusing Tyrode's buffer at a shear rate of 9000 s⁻¹ for 3 min (Supplementary Fig. 1a, b). In some experiments, platelet adhesion without thrombi was measured by incubating blood with an αIIbβ3 blocker antibody (Leo.H4, 10 μg/mL).

Platelet adhesion was monitored on recombinant mouse VWF matrix (10 μg/mL)[83] or fibrinogen matrix (100 μg/mL) after a 3 min perfusion and washing for 2 min. In some experiments, blood was incubated with actin drugs: Jasplakinolide (Jasp) (1 μM), Latrunculin A (LatA) (500 nM in Fig. 1 and 125 nM for rescue experiments) or dimethyl sulfoxide (DMSO, 1/500 vol/vol) as control. Adhesion stability was also evaluated by submitting adherent platelets to stability challenge where shear was increased to a shear rate of 9000 s⁻¹ (Supplementary Fig. 1).

All the following assays were realized on recombinant mouse VWF matrix (10 μg/mL).

Platelet velocity under flow was evaluated by perfusing blood at a shear rate of 1500 s⁻¹ in the presence of αIIbβ3 blocker (Leo.H4, 10 μg/mL)

to prevent VWF interaction with integrin αIIbβ3. Only 10% of the blood was labeled with rhodamine 6G (10 μg/mL) (Supplementary Fig. 1). In some experiments, blood was treated with actin drugs: Jasp (1 μM), LatA (500 nM in Fig. 1 and 125 nM for rescue experiments), or DMSO as control. Rolling platelets were video-recorded (1 frame/second) for 300 s for off-in analysis. Platelet velocity was then calculated by following each platelet for 40 s.

αIIbβ3 activation was measured in flow condition on adherent platelets following our standard protocol by perfusing phycoerythrin (PE)-conjugated JON/A antibody, a mAb specific for the activated conformation of mouse integrin αIIbβ3, (1/5 dilution) for 8 min and washing with Tyrode's buffer for 2 min at a shear rate of 1500 s$^{-1}$ (Supplementary Fig. 1). In rescue experiments, blood was incubated with actin drugs LatA (125 nM) or DMSO as control.

Tether dynamics was evaluated by labeling only 10% of the blood with rhodamine 6G (10 μg/mL). Blood was perfused for 3 min (Supplementary Fig. 1), platelets were observed in real-time, and images were taken every 20 s.

In all experiments, platelets were observed with an inverted epifluorescence microscope (×20; Nikon Eclipse TE2000-U). Analyses were done with MetaMorph 7.0r1 software (Molecular Devices).

### DAMI cell culture and transduction

DAMI cells (MEGAKARYOBLAST, HUMAN, CRL-9792™, ATCC, authentication by commercial source) were grown in RPMI1640 medium with Glutamax supplemented with 10% fetal bovine serum and 1% Penicillin/Streptomycin in 5% CO$_2$ condition at 37 °C. For MICAL1 silencing, DAMI cells were transduced with lentiviral particles expressing MICAL1 shRNA as previously described[30] or Luciferase shRNA as control (in pTRIP IZIE-GFP vector)[84] with a multiplicity of infection of 15. Transduction efficiency was evaluated by flow cytometry with green fluorescent protein (GFP) as reporter expression and MICAL1 protein expression was evaluated by western blotting.

### DAMI cells deformation under shear

MICAL1 and control shRNA transduced GFP-DAMI cells were adjusted in Tyrode's buffer at 4 × 106/mL, and perfused at a shear rate of 1500 s$^{-1}$ for 3 min on human VWF matrix (100 μg/mL), perfusion was then stopped to allow their adhesion for 20 min. Deformation was then induced by perfusing Tyrode's buffer and by increasing shear every 5 min from 27 to 164 dynes/cm$^2$. Images were taken every 5 min with an inverted epifluorescence microscope (×20; Nikon Eclipse TE2000-U). The shear strain was calculated by the difference of cell length in the shear direction with:

$$Strain(\%) = 100 \times [(Length\ T_x - Length\ T_0)/Length\ T_0].$$

DAMI cells' size (cell area), circularity, and strain (cell length) were measured with ImageJ 19.13.33.

### Immunofluorescence on flow samples subjected to shear

Perfusion assays were performed with blood on recombinant mouse VWF matrix (10 μg/mL) at a shear rate of 1500 s$^{-1}$ and in some experiments at a shear rate of 9000 s$^{-1}$. Coverslips were recovered at the end of flow experiments, and adhesive platelets were fixed with paraformaldehyde 4% in cytoskeleton buffer (glycerol 4 M, PIPES 0.2 M, EGTA 2 mM, MgCl$_2$ 2 mM, pH 6.9) for 20 min then permeabilized with Triton X-100 0.1% in cytoskeleton buffer for 10 min. Platelets were stained depending on the experiments with either Alexa Fluor 448-phalloidin, integrin β$_3$ or GPIbα antibodies. Slides were mounted with Prolong gold antifade and then observed by epifluorescence microscopy (x100, Eclipse Nikon 600). Pictures were analyzed with ImageJ. F-actin and GPIbα were quantified with integrated density (IntDen).

### Co-localization analysis

Flow assays were performed with blood on recombinant mouse VWF matrix (10 μg/mL) at a shear rate of 1500 s$^{-1}$. Coverslips were then recovered, and lipid rafts were labeled with cholera toxin subunit B[60]-AF555 (CT-B) (1 mg/mL) for 10 min at 4 °C before being incubated with an anti-CT-B for 15 min at 4 °C. Platelets were then fixed with 4% PFA for 15 min at 4 °C and then stained with a GPIbα antibody for 1 h at RT. Images were acquired with an inverted Nikon Eclipse Ti-E microscope equipped with a CSU-X1 spinning disk confocal scanning unit (Yokogawa) coupled to a Prime 95S scientific complementary metal-oxide semiconductor (sCMOS) camera (Teledyne Photometrics) using a X 100 1.4 NA CFI Plan APO VC objective lens and MetaMorph software. To analyze the co-localization of dual fluorescence, images were acquired at the interface between platelets and the matrix and Manders' overlap coefficients was used. Manders' coefficients were calculated using the plug-in JACoP from ImageJ and ranges from 0 (no overlap) and 1 (perfect overlap).

### F-actin content and platelet spreading

Washed platelets remained resting or were stimulated with thrombin for 10 min at 37 °C. Platelets were fixed with an equal volume of paraformaldehyde 8% in 2x cytoskeleton buffer (4 M glycerol, 0.2 M PIPES, 2 mM EGTA, 2 mM MgCl$_2$) diluted in phosphate-buffered saline (PBS), then treated with 0.1 % Triton™ X100-PBS-cytoskeleton buffer containing 10 μM phalloidin-FITC for 30 min. Samples were centrifuged for 10 min at 200 × g, resuspended in PBS and immediately analyzed with an Accuri C6 cytometer.

Coverslips were coated with fibrinogen (100 μg/mL) or recombinant mouse VWF (10 μg/mL) overnight at 4 °C. After washing twice with PBS, the coverslips were blocked with BSA (5 mg/mL) for 1 h at RT, and washed twice with PBS before use. For fibrinogen spreading, washed platelets were activated with PAR4-AP (150 μM), while for VWF, washed platelets were incubated with botrocetin (2 μg/mL) before being deposited for 30 min at 37 °C and then fixed (paraformaldehyde 4% in cytoskeleton buffer) and permeabilized with 0.1% Triton X-100 in cytoskeleton buffer. F-actin was stained with Alexa Fluor 488-labeled phalloidin (26.4 nM), and then slides were mounted with Prolong gold antifade and visualized by epifluorescence microscopy (x100, Eclipse 600, Nikon). Platelet morphology was divided into 4 states: resting, filopodia, filopodia + lamellipodia, and platelets presenting only lamellipodia. Pictures were then analyzed with ImageJ, and F-actin was quantified with integrated density (IntDen).

### Mouse platelet preparation

Blood was collected by cardiac puncture of anaesthetized mice with 80 μM D-phenylalanyl-L-prolyl-L-arginine chloromethyl ketone (PPACK) and 10% (vol/vol) ACD-C buffer (124 mM sodium citrate, 130 mM citric acid, 110 mM dextrose, pH 6.5). Washed platelets were isolated by centrifugation as previously described[85] and resuspended in Tyrode's buffer and then 2 mM Ca$^{2+}$ were added before platelet activation.

### VWF binding assays

In some experiments, mouse washed platelets were treated or not with actin drugs: Jasp (1 μM) or LatA (0.5 μM) or DMSO as control for 15 min. Platelets were treated in the presence of an αIIbβ3 blocker (Leo.H4, 10 μg/mL) to prevent VWF binding to integrin αIIbβ3, with either 5 μg/mL of botrocetin with various concentrations of recombinant mouse VWF (0–5 μg/mL) or with 5 μg/mL recombinant mouse VWF with various concentrations of botrocetin (0–10 μg/mL). Samples were then incubated for 5 min without stirring at RT. VWF binding was evaluated as previously described[86]. Samples were analyzed with an Accuri C6 cytometer (BD Biosciences).

## Human platelet preparation

Blood samples from healthy individuals were obtained from the French Blood Establishment (Etablissement Français du Sang (EFS); ref: C CPSL UNT-N°18/EFS/031), upon informed consent, in accordance with the tenets of the Declaration of Helsinki. Venous blood from healthy donors was collected in 10% (vol/vol) ACD/A buffer (75 mM trisodium citrate, 44 mM citric acid, 136 mM glucose, pH 4) for experiments with washed platelets. Platelets were washed in the presence of apyrase (100 mU/ml) and prostaglandin E1 (1 μM) to minimize platelet activation. Platelet count was adjusted in Tyrode's buffer (3 × $10^8$ platelets/mL)[87].

## Isolation and analysis of actin filaments and associated proteins

Washed platelets (3 × $10^8$/mL) were lysed with 4x PHEM buffer (PIPES 240 mM, HEPES 100 mM, EGTA 40 mM, MgCl$_2$ 8 mM, pH 6.9) containing 0.5% Triton X-100, 8 μM phalloidin and protease and phosphatase inhibitors. Triton X-100-soluble and -insoluble fractions were separated by centrifugation at 4 °C for 20 min at 15,600 × $g$[50]. Insoluble pellets were washed twice with 1x PHEM buffer with protease inhibitors. Total platelet lysates, soluble supernatant and insoluble pellets were supplemented with Laemmli buffer and subsequently subjected to SDS-PAGE.

## Co-immunoprecipitation

Human washed resting or VWF (10 μg/mL)/ristocetin (0.4 mg/mL)-treated platelets were lysed with PHEM buffer (vol/vol) containing 1% Triton X-100, 0.5% N-octylglucoside, 2 μM phalloidin as well as protease and phosphatase inhibitors. GPIbα was precipitated by incubation of the lysates with an anti-GPIbα antibody (2 μg)-coated protein G-magnetic beads, 1 h at 4 °C. Magnetic beads were washed 5 times with washing buffer (500 mM NaCl, 10 mM Tris pH 7.4, 1 mM EDTA, 1% Triton X-100, protease and phosphatase inhibitors) before 2x Laemmli buffer was added.

Mouse washed resting platelets were lysed with PHEM buffer containing 1% Triton X-100, 2% N-octyl glucoside, 2 μM of phalloidin with protease and phosphatase inhibitors.

In some experiments, immunoprecipitations were performed on shear-activated platelets. Flow assays were performed with blood on recombinant mouse VWF matrix (10 μg/mL) at a shear rate of 1500 s$^{-1}$. Coverslips were then recovered, and platelet lysate was made by scraping for 30 seconds the coated part of the coverslip (2.82 cm$^2$) with 50 μL of PHEM 1% Triton X-100 buffer, with 2 μM of phalloidin, protease and phosphatase inhibitor cocktail mixture. Platelets from two mice were pooled, and lysates were adjusted to 300 μL with the same lysis buffer. Both washed resting platelets and shear-activated platelets were subjected to immunoprecipitation. Mouse GPIbα was precipitated by incubating the lysates with anti-GPIbα antibodies (mix clone Xia.G5/Xia.G7) (2 μg)-coated protein G-magnetic beads, 1 h at 4 °C. Magnetic beads were washed 5 times with 1x lysis buffer before 2x Laemmli buffer was added. Human and mouse samples were subjected to SDS-PAGE.

## Western blotting

Immunoprecipitation samples and total platelet samples lysed in SDS denaturing buffer (50 mM Tris, 100 mM NaCl, 5 mM EDTA, supplemented with protease and phosphatase inhibitors, 100 mM dithiothreitol, pH 7.4) were subjected to SDS-PAGE and transferred to nitrocellulose or were analyzed on an automated capillary-based immunoassay platform; Wes (ProteinSimple). The membranes were incubated with various primary antibodies (Supplementary Table 1). Immunoreactive bands were visualized using Enhanced Chemiluminescence Detection Reagents (Pierce) or fluorescent antibodies. Images of the chemiluminescent signal were captured using G:BOX Chemi XT16 Image Systems and quantified using Gene Tools version 4.0.0.0 (Syngene) or with the Amersham Typhoon and quantified using ImageQuant TL image analysis software (GE Healthcare Bio-sciences AB). Unedited blots are provided in Source data File.

## Transmission electron (TEM) and scanning electron (SEM) microscopies

TEM: Washed platelets were fixed by incubation for 1 h at room temperature with 1.25% glutaraldehyde in 0.1 M phosphate buffer, pH 7.2, centrifuged for 10 min at 1100 × $g$, and washed once in phosphate buffer. Platelets were kept in 0.2% glutaraldehyde at 4 °C until processing for standard transmission electron microscopy analysis of platelet morphology, as described previously[88]. Sections were examined using a JEOL JEM1400 transmission electron microscope with a Gatan Orius 600 camera and Digital Micrograph software (Lyon Bio-Image, Centre d'Imagerie Quantitative de Lyon Est). Staining of the OCS was done with ruthenium red of surface-connected membranes as described by Mountford et al.[6].

SEM: After flow challenge at a shear rate of 9000 s$^{-1}$, platelets were fixed by incubation for 24 h in 2% glutaraldehyde in 0.1 M phosphate buffer (pH 7.2). Samples were then washed in PBS and post-fixed by incubation with 2% osmium tetroxide for 1 h. Platelets were then fully dehydrated in a graded series of ethanol solutions and dried in hexamethyldisilazane. Finally, samples were coated with 40 Å platinum, using a GATAN PECS 682 apparatus (Pleasanton), before observation under a Zeiss Ultra plus FEG-SEM scanning electron microscope (Oberkochen).

## Flow cytometry analysis

Surface glycoproteins expression and surface β-galactose were measured in diluted whole blood (1/20 with Tyrode's buffer) using appropriate fluorophore-conjugated antibodies for GPIbα, GPIbβ, GPIX, integrin αIIbβ3, integrin α2, GPVI or *Ricinus communis* Agglutinin (RCA-I) lectin (12.5 μg/mL). Platelet activation was evaluated in washed platelets (3 × $10^8$/mL) with a range of several agonists for 10 min without stirring at room temperature (RT). Activation level was evaluated by using JON/A (active conformation of mouse integrin αIIbβ3), and by measuring the P-selectin exposure. Platelets were incubated with antibodies or RCA-1 lectin for 20 min at RT and then diluted with PBS. Acquisition was performed with 10,000 events in the platelet gate. DAMI cells, $10^6$ cells were centrifuged and resuspended in 50 μL de buffer A (0.5% BSA, 2 mM EDTA in PBS, pH 7.2) and appropriate fluorophore-conjugated antibodies. Cells were incubated 10 min at 4 °C before being washed with buffer A and centrifuged, and the pellet was resuspended with 600 μL of buffer A. Acquisition was performed with 20,000 events in the gate of living cells. Samples were collected with an Accuri C6 cytometer and analyzed with C6 plus analysis software (BD Biosciences). Gating strategy is provided in Supplementary Fig. 10.

## In vivo platelet half-life

Platelet lifespan was determined by intravenous injection of DyLight488-labeled anti-GPIX antibody (5 μg; X488, Emfret Analytics) into 8-week mice. Blood samples were collected into PBS-EDTA 0.5 M tubes at various time points after injection. The percentage of marked platelets was determined by flow cytometry (Accuri C6 cytometer).

## Bone marrow megakaryocyte

Frozen sections (5 μm) were fixed with acetone at −20 °C for 10 min, rinsed in PBS and treated with 1% BSA and permeabilized with 0.1% Triton X-100, then incubated in the oven for 1 h at 37 °C with the primary antibody FITC rat anti-mouse CD41, as well as the rabbit anti-laminin antibody. The slides are washed twice and incubated for 1 h with the Texas Red secondary antibody goat anti-rabbit.

After two additional washes, the slides are stained and mounted with VECTASHIELD® Antifade Mounting Medium with DAPI before being stored in the dark at 4 °C. Samples were analyzed using confocal

microscopy (Confocal Laser Scanning Microscope, CLSM, Leica TCS SP5, Leica Biosystems France) and image analysis software (ImageJ).

## Platelet aggregation

Mouse platelet aggregation was monitored by measuring light transmission through the stirred suspension of washed platelets ($3 \times 10^8$/mL) at 37 °C using a Chronolog aggregometer model 700 (Chrono-log Corporation). Several agonists triggered platelet aggregation. Results are expressed as the percentage change in light transmission with respect to the blank (buffer without platelets), set at 100%. In some experiments, human platelets were activated with Ristocetin (0.4 mg/mL) in the presence of VWF (Wilfactin®, 10 µg/mL) in stirring conditions. Data were collected with AGGRO/LINK®8 software.

## Evaluation of ATP release

Samples of platelet secretion were obtained after aggregation triggered by 2 different concentrations of thrombin (40 and 100 mU/mL) after incubation with Jasplakinolide (Jasp; 1 µM) or DMSO as control. EDTA (15 mM) was added to samples before centrifugation at $6600 \times g$ for 1 min. The supernatant was then used to measure ATP secretion with a luciferin-luciferase detection kit (Molecular Probes A22066). Light emission was assessed with a luminometer (Fluoroskan Ascent FL; Thermo LabSystems).

## Statistics

All displayed values are mean ± standard deviation (SD). Significances were calculated using GraphPad Prism 10.5.0 (Dotmatics, USA). The following tests were used as indicated: paired and unpaired $t$-tests (Two-tailed), one-way or two-way analysis of variance (ANOVA) followed by a post hoc test for multiple comparisons, as indicated in the figure legends, Pearson correlation coefficient, and linear regression. In all tests, $p$-values > 0.05 were considered as not significant (ns). Exact $p$-values are indicated in the figure and/or in legend. Replicates and $F$ values and degrees of freedom (df) for ANOVA and $t$-values and df for $t$-tests are indicated in the figure legend. Data are represented in histograms, violin plots or curves.

## Reporting summary

Further information on research design is available in the Nature Portfolio Reporting Summary linked to this article.

# Data availability

The authors declare that the main data generated in this study supporting the findings are available within the article and the main data are provided in the Supplementary Information and Source Data files. Source data are provided with this paper.

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

## Acknowledgements

This manuscript is in honor of Dr. Jean-Claude Bordet, who dedicated his research to the electron microscopy notably presented in this paper. He passed away in 2024. We thank the Institut Biomédical du Val-de-Bièvre (IBVB-UMS44), for technically and facility supports and the "animal facility Pincus" UMS44, University of Paris Saclay and their zootechnicians, the "Cellular and Molecular Imaging Facility", US25 INSERM, UAR3612 CNRS, Faculty of Pharmacy, Université of Paris-Cité for technical support. We also thank Eloïse Pascal (INSERM U1176) for her excellent technical assistance with the carotid model. We thank Isabelle Dusanter-Four and Amandine Houvert (INSERM U1016) for lentiviral particles production. This work was supported by INSERM, CNRS, University Paris-Saclay, the Institut Pasteur and French Research Agency (ACTOMIC: ANR-22-CE14-0013-01 grant to A.K. and A.E., TRIMEP: ANR-17-CE18-0025 grant to A.K. and D.B., MACROIT: ANR-22-CE17-0043-03 grant to A.K. and H.R.).

## Author contributions

J.S. carried out the experiments presented in Figs. 1a–d, 2a, 3g, h, j–l, 4, 5, 6, 7a–c, f–l, 8 and Supplementary Figs. 1a,b, 3, 4, 5c, 6a–d, 7a–d, 9 and Tables 1 and 2 and Supplementary Movies; C.S. the experiments in Figs. 1e, f, 2b, e, f, 3g, i, 7d, e, g, h and Supplementary Figs. 1c, 2b, 5a, b, d, e, 8 and Tables 1 and 2; S.F. the experiments in Fig. 8; C.R. the experiments in Supplementary Table 1 and Supplementary Fig. 5d, e; S.E. and C.N. performed scanning microscopy experiments in Fig. 6g, h; E.R. performed the experiments in Supplementary Fig. 2c; J.C.B performed TEM experiments in Fig. 3b–f and Supplementary Fig. 7e; P.M. and Y.B. expert advices in the in vivo experiments in Fig. 3 and flow in Fig. 4; R.J.P provided *Mical1*^lox/lox mice; A.K. Figures 2a, c, d, 3a; Supplementary Data 2; Supplementary Tables 1 and 2; Supplementary Figs. 2a,c; H.R. expert advices experiments in Supplementary Fig. 2; C.V.D. provided recombinant murine VWF; J.S., A.E, A.K. scheme in Fig. 9; A.K. conceived the project; A.K., A.E., and F.A. oversaw all experiments, designed and interpreted the experiments; J.S., A.K., F.A., A.E. and S.F. wrote the manuscript; H.R., D.B., A.E., A.K. secured funding; ANR ACTOMIC to A.K. and A.E. All authors approved the final version of the article.

## Competing interests

The authors declare no competing interests.
