## [Transparent Peer Review file · Nature Communications]

F-actin disassembly by the oxidoreductase MICAL1 promotes mechano-dependent VWF-GPIIb/IIIa interaction in platelets

Corresponding Author: Dr Alexandre Kauskot

Version 0:

Reviewer comments:

Reviewer #1

(Remarks to the Author)

The manuscript "Mechano-dependent VWF-GPIIb/IIIa interaction in platelets requires F-actin disassembly by the oxidoreductase Mical1" by Solarz et al. investigates the role of actin cytoskeleton dynamics in the mechano-dependent interaction between von Willebrand factor (VWF) and the platelet receptor GPIIb/IIIa. The authors identify Mical1 as a key enzyme mediating actin disassembly under shear conditions, thereby contributing to GPIIb/IIIa-signaling and reinforcing platelet adhesion to VWF. The study presents novel insights into the regulation of platelet adhesion and thrombus stability, linking actin oxidation to platelet function.

Overall, the manuscript is well-written and presents interesting findings that contribute to our understanding of platelet mechanotransduction. The study is of clear interest to the field, and most conclusions are backed up by compelling experimental evidence. However, there are several points that require further clarification or additional discussion before the manuscript is suitable for publication. Below, I outline some key aspects that should be addressed to strengthen the study.

Major comments:

- 1) Standard deviation (SD) represents the variability within a sample, while the standard error of the mean (SEM) quantifies the precision of the sample mean estimate. Given the rather low sample size in this study, SEM might not be the most appropriate choice, as it can give a misleading impression of variability, particularly in biological experiments where individual variation is relevant. SD would better reflect the true distribution of the data and should be reported alongside SEM, if needed.
- 2) Page 5, line 19f.: "suggesting that a mechano-dependent signaling downstream of the VWF-GPIIb/IIIa interaction leading to platelet adhesion is impaired upon F-actin stabilization" – I am not convinced that this necessarily implies signaling. A defective actin stabilization could simply result in weaker receptor anchoring, independent of classical signaling cascades. While the outcome is the same, the data seem to support a role for Mical1 as an effector rather than as a signaling molecule. Along the same lines, wouldn't a more stable actin cytoskeleton (as in Fig. 2C) generally stabilize interactions between cytoskeleton-associated proteins and receptors?
- 3) Mical1-deficient platelets display hyper-aggregation in response to most agonists, which the authors attribute to enhanced dense granule secretion. Have the authors tested aggregation in the presence of high-dose apyrase (to remove ATP) to validate this interpretation?
- 4) The observed increase in dense granule secretion is interesting, but a mechanistic discussion is missing. What could explain this phenotype mechanistically?
- 5) The larger thrombus size in Mical1-deficient blood (Fig. 4a) is surprising. Have the authors tested whether inhibiting second-wave mediators would normalize thrombus formation? This could clarify whether the phenotype is indeed due to elevated dense granule secretion.
- 6) I agree with the authors, that vWF is captured by collagen and contributes to thrombus formation by enabling platelet tethering. However, GPVI is the key receptor driving thrombus formation on collagen and I assume that blocking GPVI would

abrogate thrombus formation on collagen in this experimental setting. Thus, GPVI should at least be mentioned and the contribution of GPVI should be discussed regarding Fig. 4.

7) I am aware that studying GPIIb-mediated effects is notoriously challenging as it always requires shear forces. However, I am bit skeptical regarding Fig. 5c as KO platelets appear to adhere less, and the microaggregates seem smaller. Was JON/A-PE MFI assessed per visual field or normalized per platelet? Only the latter would be valid. In any case, the stronger evidence that Mical1 contributes to GPIIb-driven integrin activation is presented in Fig. 7e (higher rolling velocity). The authors might consider moving Fig. 7e to Fig. 5. Along the same lines, did they quantify the percentage of rolling platelets on VWF in WT and Mical1-deficient platelets?

8) I am not convinced that the lipid raft part is backed up sufficiently by experimental data.

a) The literature cited to introduce the concept is relatively outdated, and it is unclear whether modern techniques would still support these claims.

b) The resolution of the images in Fig. 7i & 7j does not seem sufficient to assess co-localization. The staining appears largely homogeneous, making it difficult to draw firm conclusions.

c) The observed difference in the Manders' coefficient could be attributed to differences in tether formation frequency rather than true changes in lipid raft localization (as tethers appear cholera toxin-negative). What happens if only the platelet body is considered?

d) Cholesterol depletion with methyl- β -cyclodextrin (Supplemental Figure 8) affects the entire plasma membrane, not just lipid rafts. The claim that "lipid raft disruption inhibits platelet adhesion on VWF" seems overstated.

Minor comments:

1) Western blot images should indicate molecular weights.

2) Clarification for Figure 1a: What does "platelet adhesion in %" refer to? Is it the fraction of total platelets adhering or the surface coverage? If the latter, the reported percentages seem rather low.

3) Figure 3d: Were the results obtained after 3 or 6 minutes?

4) Rephrase page 6, line 14: Fig. 4a refers to ex vivo thrombus formation. The current phrasing "in Mical1^{-/-} mice" is misleading.

5) Piezo1 as a potential mechanosensitive factor: The authors performed a comprehensive analysis of actin regulators (Supplemental Figure 5), but did they assess whether Mical1 affects Piezo1 expression? This could provide an additional link to the observed defects in mechanotransduction.

6) Please indicate flow direction in Figure 6.

Reviewer #2

(Remarks to the Author)

It has been well established that actin dynamics in mammalian cells is controlled by monooxygenase activity of MICAL enzymes that oxidize specific actin methionine residues which results in actin filament disassembly. In this study, Mical1 has been linked to actin filament disassembly in platelets mediated by the VWF-GPIIb interaction under high fluid shear. The studies have been well designed, the data is robust and the results support the conclusions.

Comment

Platelet interactions during the developing thrombus are dynamic where they can undergo several cycles of adherence, rolling and/or detachment. The studies are presented as a linear sequence of irreversible events, whereas all steps are potentially (likely) reversible. Notably, actin oxidation by MICALs has been shown to be reversed by methionine sulfoxide reductases (MSR) in other cells. It is plausible that actin filament disassembly in platelets is a balance between MICAL and MSR activities at any stage.

Reviewer #3

(Remarks to the Author)

The manuscript by Solarz et al. studied the role of Mical1, an enzyme involved in the disassembly of F-actin, and subsequent platelet adhesion and stability in response to high shear flow. The authors examine the role of actin polymerization and depolymerization with actin drugs and a Mical1 knockout mice used for in vivo and in vitro experiments. In the presence of shear flow, they found that F-actin disassembly associated with the GPIIb-IX-V complex promotes platelet adhesion and stability, while F-actin assembly has the opposite effect. Additionally, the data suggests that Mical1 enhanced platelet adhesion, thrombus stability, and occlusion at the lesion site. Without Mical1, embolization was more likely to occur. Additionally, Mical1^{-/-} platelets were shown to have morphological changes, including elongated membrane tethers and increased F-actin levels, and decreased localization of GPIIb in lipid rafts. The authors concluded that Mical1 regulates shear-dependent actin reorganization within the GPIIb-IX-V complex, which plays a role in platelet adhesion and stability.

While a strong manuscript, this reviewer has concerns about overstatements in the title and text, additional conditions to support the author's conclusions, and characterization of the Mical1 KO.

MAJOR COMMENTS

1. Title. The title of the manuscript is "Mechano-dependent VWF-GPIIb interaction in platelets requires F-actin disassembly by the oxidoreductase Mical1." This reviewer is convinced that Mical1 is involved but sees "require" as an overstatement given that Mical1^{-/-} platelets still bind to VWF under shear. This reviewer suggests the authors modify the title and soften the "require".

2. LatA and Jasp characterization.

a. From the results in Fig. 2b, the authors claim that "the redistribution of Mical1 and GPIIb from the soluble to the insoluble cytoskeletal fraction was abolished upon F-actin depolymerization by LatA". However, this reviewer sees a distinct band in the blot image for Mical1 +LatA at 2 mins, which does not seem to be represented in the graphs or support the "abolished" claim. Additionally, error bars are missing (or perhaps are too small to see) for several of the data points in Fig. 2b. To make the conclusion more convincing, the authors could consider a combination of the following approaches (1) include a table with the raw western blot values, (2) plot the +LatA lines on a separate graph with a smaller scale to show the true signal and error bars, (3) demonstrate the effect on the Triton X-100 insoluble fraction when F-actin is stabilized with +Jasp.

b. In supplemental Figure 1b, the authors show that LatA results in destabilization of F-actin and that Jasp results in stabilization of actin. However, this conclusion and effect size is based on N = 1 with no statistics. An estimate of the effect size at the chosen concentrations is important to the conclusions of this work; N = 1 with no statistics is insufficient.

c. The conclusion that Mical1 disassembles F-actin and enables GPIIb translocation into lipid rafts is critical to this paper (Figure 7). This conclusion would be strengthened if both Mical1^{+/+} and Mical1^{-/-} were treated with LatA and Jasp and GPIIb/lipid raft was measured. The condition that is especially important is Mical1^{+/+} platelets treated with Jasp.

3. Mical1^{-/-} characterization.

a. In Figure 3a, it appears that the Mical1^{-/-} has substantially more alpha-tubulin. In supplemental figure 5, it appears that there is no difference in alpha-tubulin. This reviewer suggests that the authors provide some context, explanation, and/or additional evaluation.

b. The authors find that mean platelet volume is not significantly different with and without Mical1 (Supplementary Table 2), but that platelet surface area is significantly higher in the absence of Mical1 ($p < 0.001$, Figure 3b). This is a key point that should be further investigated. Either (1) the platelet shape is different in the absence of Mical1, such that it is possible to have a different surface area but the same volume OR (2) the volume per platelet is different, but the difference is not identifiable when comparing average platelet volume per mouse instead of examining platelet volume on a per platelet basis (similar to what is done in Figure 3b). Based on the image in Figure 3c, the higher surface area of Mical1^{-/-} platelets, and the trending (though not significantly lower) MPV of Mical1^{-/-} platelets, this reviewer suspects that platelets lacking Mical1 have a significantly more oblong shape.

c. In supplementary Figure 2c, the authors find that Mical1 deficiency does not affect number of MKs in the bone marrow based on N = 3 mice. This reviewer is not convinced of this conclusion. Estimating from the graph, the number of MK per square mm with Mical1 is 17 and the number without is 12 (estimated 1.4-fold difference). This is a similar effect size to what is observed in carotid artery occlusion time (Figure 3f), yet is ns due to the lower quantity of mice characterized. The authors should provide a compelling rationale regarding why N = 3 was sufficient here (but not elsewhere throughout the paper) or should conduct additional experiments to better characterize this apparent 1.4-fold difference.

4. Overstatement in main text.

a. In a subtitle of the main text and figure title of Fig. 3, the authors summarize their findings as "Mical1 is required for hemostasis and thrombosis in vivo". This reviewer sees this as an overstatement not sufficiently supported by the data. However, this reviewer would agree that the data indicates Mical1 contributes to hemostasis and thrombosis in high shear conditions. For example, hemostasis is reached in Figure 3e, albeit with a longer bleeding time, even in the absence of Mical1. Also, Fig. 3h does not seem to support this claim without indicating 'high shear conditions'. Further, this reviewer suggests adding more as to why there are differences in the effects of Mical1 in the carotid artery, mesenteric arteriole, and mesenteric venule.

b. In Supplementary Fig. 3a, b this reviewer does not agree that the data fully support the claim that "agonist-induced platelet aggregation was significantly enhanced in Mical1^{-/-} platelets due to increased release of dense granules". For several concentrations of agonists (Supplementary Fig. 3a), there is no significance between Mical1^{+/+} and Mical1^{-/-} and ADP shows no significance between aggregation for the two conditions. Only Supplementary Fig. 3b seems to show enhanced dense granule release via ATP concentration in response to thrombin.

c. In Fig. 5f, 5g, the authors conclude that "Mical1 deficiency induces excessive F-actin polymerization"; however, this reviewer sees this conclusion as a stretch for the data shown. The figure only shows the defects in the adhesion and stability of Mical1^{-/-} platelets. This statement is better supported by the results shown in Fig. 6.

MINOR COMMENTS

1. Abstract. In the abstract, the authors state, "but whether actin dynamics is important for platelet function under hemodynamic, high shear conditions, is largely unknown". This reviewer sees this statement on the gap in the field as an overstatement. Actin dynamics have been characterized in many conditions under shear such as platelet shape change and spreading, granule secretion, clot retraction and stability, mechanosensing, and force transmission. The authors should rephrase this statement to more specifically state the gap this manuscript fills.

2. Misleading statements about VWF exerting force. Overall, the introduction is well-written and appropriately cites the existing literature. One constructive comment is that several times, the authors mention that the force between VWF and GPIIb is induced by VWF pulling (introduction line 8, line 19). This statement is misleading. VWF is not alive and does not pull (on its own). It is this reviewer's understanding that tension on the VWF-GPIIb bond comes from two main sources: 1) blood flowing over the platelet-VWF bond, pulling on both VWF and platelets and 2) platelet contraction. This reviewer strongly suggests clarifying this point to better frame the paper and to dispel the misunderstanding that VWF pulls on the

bond.

3. Experimental details. This reviewer suggests

a. Mentioning that mouse blood was used within the main text for experiments in figure 1. This information it is currently only in the figure caption.

b. Explaining what is meant by “independent experiment.” It is unclear whether an ‘independent experiment’ is a different mouse on a different surface on a different day or whether an ‘independent experiment’ may be the same mouse blood on the same day, but only a different surface. E.g., in the captions of Figure 1a-c and Figure 4a-c.

c. Including many fields of view were captured in each figure. This is done in Figure 5c, but is not done in consistently through the paper (e.g., Figure 1, 4, 5).

d. Providing a rationale for revering to shear in terms of s^{-1} and then switching to dynes/cm² in supplementary figure 6

e. Identifying a direction of flow on the images in figures 6 and 7.

4. Experimental rationale. This review suggests including

a. The rationale for the 1500 s^{-1} is that this is an arteriolar shear rate. This reviewer suggests adding a rationale about why the higher shear rate of 9000 s^{-1} was selected. The authors may find this publication helpful

10.1182/bloodadvances.2022007550. It would also support the manuscript to provide estimated shear rates for the conditions in Figure 3e-h, either by experimental measurement or estimations from the literature.

b. The rationale for how the authors determined the number of biological or technical replicants ($n = ?$) to include in each study.

c. The rationale for why the concentrations of Jasp1 and LatA were selected.

5. Supplementary data. This reviewer commends the authors for providing substantial supplementary information, such as precise product information. This reviewer additionally suggests that the authors add the following to the supplementary data:

a. a supplementary figure with representative raw flow cytometry data used to create Figure 1d,

b. supplementary videos of raw data used to create Figure 1c,

c. supplementary figures of unedited (i.e., no repositioned lanes) blot images for all blots,

d. supplementary videos of raw data used to create Figure 3i,

e. supplementary videos of raw data used to create Figure 4a-c,

f. supplementary videos of raw data used to create Figure 5,

g. supplementary videos of raw data used to create Figure 6a-f,

h. supplementary videos of raw data used to create Figure 7e, 7h

i. Raw data of thrombus size after stability challenge in Figure 4b and Figure 5b (in addition to the existing quantification of thrombus instability)

6. Data presentation. Figure 7i and j use green and red immunofluorescence images and an overlay. Given that red/green colorblindness is the most common type of colorblindness, this reviewer suggests that the authors choose different colors. There are many alternative colorblind-friendly options.

7. Typos, grammar, and clarifications.

a. The capitalization of MICAL1 is inconsistent. The manuscript uses both “MICAL1” and “Mical1”.

b. Supplementary table 2 is titled “... Mice separated by gender.” “Male” and “female” are terms to describe sex, not gender. The term “gender” should be changed to “sex”.

c. Page 7, line 27-28 states, “we therefore hypothesized that this embolism could be due to impaired adhesion...” Given that the authors showed that the KO has more platelet adhesion under low shear, the reviewers suggest that perhaps the authors mean, “we therefore hypothesized that this embolism could be due to impaired adhesion in high shear conditions...”

d. This reviewer encourages the authors to expand upon the translational implications of Mical1 in patient care.

Reviewer #4

(Remarks to the Author)

Reviewer #5

(Remarks to the Author)

Version 1:

Reviewer comments:

Reviewer #1

(Remarks to the Author)

My comments have been addressed and I congratulate the authors on their work.

Reviewer #2

(Remarks to the Author)

The authors have adequately addressed my comment in the revision.

Reviewer #3

(Remarks to the Author)

The authors were responsive to reviewer suggestions. I particularly appreciate the added shape analysis and the lipid raft/GPIb colocalization analysis; both of these analyses (and others) were valuable additions.

Reviewer #4

(Remarks to the Author)

Reviewer #5

(Remarks to the Author)

REVIEWER COMMENTS

We would like to thank the reviewers for their thoughtful and positive feedback on our manuscript. We are also grateful for their constructive comments, which have helped us to improve the manuscript further. As detailed below, we have carefully addressed all the points raised in order to enhance the clarity and impact of our study. New supporting data have been added to new Fig. 2, 3, 5, 7 and 8 and new supplemental Fig. 1, 2, 3, 5 and 9.

Reviewer #1 (Remarks to the Author):

The manuscript "Mechano-dependent VWF-GPIIb α interaction in platelets requires F-actin disassembly by the oxidoreductase Mical1" by Solarz et al. investigates the role of actin cytoskeleton dynamics in the mechano-dependent interaction between von Willebrand factor (VWF) and the platelet receptor GPIIb α . The authors identify Mical1 as a key enzyme mediating actin disassembly under shear conditions, thereby contributing to GPIIb α -signaling and reinforcing platelet adhesion to VWF. The study presents novel insights into the regulation of platelet adhesion and thrombus stability, linking actin oxidation to platelet function. Overall, the manuscript is well-written and presents interesting findings that contribute to our understanding of platelet mechanotransduction. The study is of clear interest to the field, and most conclusions are backed up by compelling experimental evidence. However, there are several points that require further clarification or additional discussion before the manuscript is suitable for publication. Below, I outline some key aspects that should be addressed to strengthen the study.

Major comments:

1) Standard deviation (SD) represents the variability within a sample, while the standard error of the mean (SEM) quantifies the precision of the sample mean estimate. Given the rather low sample size in this study, SEM might not be the most appropriate choice, as it can give a misleading impression of variability, particularly in biological experiments where individual variation is relevant. SD would better reflect the true distribution of the data and should be reported alongside SEM, if needed.

Thank you for this valuable insight. As suggested, we have revised the manuscript and replaced SEMs with SDs in order to better represent the true distribution of the data.

2) Page 5, line 19f.: "suggesting that a mechano-dependent signaling downstream of the VWF-GPIIb α interaction leading to platelet adhesion is impaired upon F-actin stabilization" – I am not convinced that this necessarily implies signaling. A defective actin stabilization could simply result in weaker receptor anchoring, independent of classical signaling cascades. While the outcome is the same, the data seem to support a role for Mical1 as an effector rather than as a signaling molecule.

We agree with the reviewer.

We have now revised the sentence as follows (p. 5):

"Thus, F-actin depolymerization favors VWF-GPIIb α interactions, whereas F-actin stabilization is detrimental for these interactions under high shear. In contrast, in the absence shear (static condition), F-actin stabilization did not alter VWF binding to platelets (Fig. 1d). This suggests that mechano-dependent binding of GPIIb α with associated proteins and/or signaling downstream of the VWF-GPIIb α interaction important for platelet adhesion is impaired upon F-actin stabilization".

Along the same lines, wouldn't a more stable actin cytoskeleton (as in Fig. 2C) generally stabilize interactions between cytoskeleton-associated proteins and receptors?

Indeed, a more stable actin cytoskeleton could stabilize the interactions between cytoskeleton-associated proteins and receptors. Consistent with this idea, we observed that MICAL1 —an actin

cytoskeleton-associated protein— loses its association with the GPIb receptor upon F-actin depolymerization by LatA treatment (Fig. 2).

However, while a more stable actin cytoskeleton could indeed stabilize interactions between cytoskeleton-associated proteins and receptors, the dynamic nature of actin is equally important in facilitating the rapid changes necessary for proper platelet function and thrombus formation. Here, we demonstrate that the stabilization of F-actin within the GPIb complex, caused by either the absence of MICAL1 or Jasplakinolide treatment, impairs platelet interaction and adhesion (Fig. 5 and 7). Thus, an optimal level of F-actin is required for proper GPIb/VWF interactions, especially under high shear.

In the new Discussion (p. 12), we have now included this notion and added a new reference (new REF#62):

“...we showed that an appropriate level of F-actin is essential for proper platelet interaction with VWF, specifically under high shear and F-actin levels must be tightly regulated for optimal stable platelet adhesion (Fig. 1) notably to control interactions between cytoskeleton-associated proteins and receptors⁶².”

3) Mical1-deficient platelets display hyper-aggregation in response to most agonists, which the authors attribute to enhanced dense granule secretion. Have the authors tested aggregation in the presence of high-dose apyrase (to remove ATP) to validate this interpretation?

We thank the reviewer for raising this important question. To validate our interpretation, we conducted the suggested experiment and found that a high-dose apyrase restored normal aggregation levels in MICAL1-deficient platelets, which were comparable to those of wild-type platelets. These results are now included in the new Supplementary Fig. 3d:

Supplementary Figure 3: (d) Aggregation of washed *Mical1*^{+/+} and *Mical1*^{-/-} platelets activated with thrombin 600 mU/mL after incubation with 2 U/mL of apyrase or not (mean ± SD, in absence of apyrase N = 3, in presence of apyrase N = 6, multiple unpaired Student t-test, t: without apyrase (-) = 0.3269, with apyrase (+) = 0.2774, df = 14). Left panel: representative aggregation traces.

Moreover, ADP alone, a weak agonist, did not induce the secretion of dense granules in washed platelets and no difference in ADP-induced aggregation was observed between WT and MICAL1 KO platelets, as shown in Supplementary Fig. 3b.

Supplementary Figure 3b : Aggregation of washed *Mical1*^{+/+} and *Mical1*^{-/-} platelets activated with different agonists (ADP: mean ± SD, N = 3, multiple unpaired Student's t-test, t: 5 μM = 0.08071, 10 μM = 0.4036, df = 8) (PAR4-AP: mean ± SD, N: 50 μM = 4; 75 μM = 3, multiple unpaired Student's t-test, t: 50 μM = 3.256, 75 μM = 0.04475, df = 10) (Thrombin: mean ± SD, N: 30 mU/mL = 3; 40 and 80 mU/mL = 4, multiple unpaired Student's t-test, t: 30 mU/mL = 3.122, 40 mU/mL = 5.730, 80 mU/mL = 0.1483, df = 14) (Convulxin: mean ± SD, N: 150 pM = 3, 200 pM = 4; * p < 0.01, 400 pM = 3, multiple unpaired Student's t-test, t: 150 pM = 4.216, 200 pM = 3.230, 400 pM = 0.4864, df = 14). Bottom panels: representative aggregation traces.

Together, these findings support our interpretation that the enhanced aggregation in MICAL1-deficient platelets is primarily due to increased dense granule secretion, rather than a direct effect on the aggregation machinery.

We have modified the sentence in the Results on p. 7 as follows:

“Similarly, platelet aggregation and ATP secretion were significantly enhanced in *Mical1*^{-/-} platelets induced by Thrombin, Cvx, PAR4-AP agonists leading to secretion (strong agonists) but not by ADP (weak agonist) (Supplementary Fig. 3b). In line with the observed ex vivo thrombus formation, treatment with apyrase normalized the aggregation of *Mical1*^{-/-} platelets to the level observed *Mical1*^{+/+} platelets. These results indicate that the aggregation enhancement is primarily due to increased release of dense granules (Supplementary Fig. 3c, d).”

4) The observed increase in dense granule secretion is interesting, but a mechanistic discussion is missing. What could explain this phenotype mechanistically?

It has previously been demonstrated that F-actin promotes secretion and aggregation. Furthermore, the secretion is regulated by actin dynamics, by actin-binding proteins (ABPs) such as gelsolin and also involves interactions of F-actin with filamin A as well as with SNARE proteins (new references # 65-69). Since MICAL1 disassembles F-actin, MICAL1 KO increases F-actin levels which likely explains the observed increase in dense granule secretion (and aggregation). Consistently, we found that low doses of Jasplakinolide (which induces F-actin stabilization) enhanced both platelet aggregation and secretion. We have not added these new data since the main phenotype observed *in vivo* following MICAL1 depletion is a delay of occlusion in arteries (and not an enhancement of the thrombus formation). However, we can include this information if the reviewers think that it is necessary.

To provide a mechanistic explanation that supports the secretion phenotype, we have added four new references (p. 13):

“At the cellular level, our functional assays revealed three functions of MICAL1 in platelets through its role in actin disassembly. Firstly, MICAL1 plays a negative role in platelet secretion and aggregation (Fig. 4 and Supplementary Fig. 3), consistent with a role of F-actin and its dynamics in promoting

secretion and aggregation⁶⁵⁻⁶⁹. Of note, this cannot explain the defects observed *in vivo* in *Mical1*^{-/-} mice, which emphasize a dominant role of MICAL1 in shear-dependent adhesive and thrombotic functions.”

5) The larger thrombus size in *Mical1*-deficient blood (Fig. 4a) is surprising. Have the authors tested whether inhibiting second-wave mediators would normalize thrombus formation? This could clarify whether the phenotype is indeed due to elevated dense granule secretion.

As requested, we investigated thrombus formation under flow conditions in the presence of apyrase. Similarly to aggregation, the apyrase treatment restored thrombus formation in MICAL1-deficient blood to levels comparable with those in WT blood. Therefore, the enhanced thrombus formation is due to the increased secretion in MICAL1-deficient platelets.

These data have been included in the new Supplementary Fig. 3a, along with the following sentence (p. 7):

“ In particular, individual thrombi were larger in the absence of MICAL1, as evidenced by increased thrombus volume (Fig. 4a), likely due to increased dense granule secretion since thrombus size was normalized after inhibiting second-wave mediators (ADP/ATP by apyrase) (Supplementary Fig. 3a).”

New Supplementary Figure 3: MICAL1 deficiency increases platelet aggregation in a secretion-dependent manner

(a) Rhodamine 6G stained *Mical1*^{+/+} or *Mical1*^{-/-} platelets in whole blood treated with 5 U/mL of apyrase were perfused on type I collagen matrix. Thrombus size at a shear rate of 1,500 s⁻¹ (mean fluorescence intensity (MFI) ± SD, N = 5 independent experiments; two-tailed paired Student’s t-test, t = 0.04228, df = 8). Left panel: representative images.

6) I agree with the authors, that vWF is captured by collagen and contributes to thrombus formation by enabling platelet tethering. However, GPVI is the key receptor driving thrombus formation on collagen and I assume that blocking GPVI would abrogate thrombus formation on collagen in this experimental setting. Thus, GPVI should at least be mentioned and the contribution of GPVI should be discussed regarding Fig. 4.

We agree with the reviewer, and we have added this information to the Results section (p. 8):

“We therefore hypothesized that this embolism could be due to impaired adhesion in high shear conditions, reflecting defective interactions between MICAL1-deficient platelets with collagen fibers through GPVI receptors and with the VWF bound to collagen through GPIIb receptors.”

Of note, we observed the impaired adhesion phenotype becoming more pronounced as the shear rate increased. Specifically, we found no increase in occlusion time at low shear in the mesenteric venules, a significant 1.4-fold increase in the carotid, and a notable > 2-fold increase in the mesenteric arterioles. These results suggest that MICAL1 is more likely involved in the GPIIb/VWF axis rather than in the collagen/GPVI axis.

We thus have added this sentence in the Discussion (p. 12):

“We observed that the embolization phenotype became more pronounced with increasing shear rates. Specifically, no phenotype was detected in mesenteric venules, which are characterized by low shear rates, while a significant increase in occlusion was observed in mesenteric arterioles, where shear rates are higher. This shear-dependent phenotype suggests a preferential involvement of the VWF/GPIIb axis, known to be activated under high shear conditions, rather than the collagen/GPVI axis, which typically operates under low to moderate shear conditions. The absence of a phenotype at low shear further supports the idea that MICAL1 functions within a shear-sensitive pathway that relies on VWF-GPIIb interactions”.

7) I am aware that studying GPIb-mediated effects is notoriously challenging as it always requires shear forces. However, I am bit skeptical regarding Fig. 5c as KO platelets appear to adhere less, and the microaggregates seem smaller. Was JON/A-PE MFI assessed per visual field or normalized per platelet? Only the latter would be valid. In any case, the stronger evidence that *Mical1* contributes to GPIb-driven integrin activation is presented in Fig. 7e (higher rolling velocity). The authors might consider moving Fig. 7e to Fig. 5.

In the first version of the manuscript, we measured the MFI per visual field. Taking the reviewer's advice into account, we have now normalized the JON/A staining per platelet. This intensity was calculated by measuring 600 platelets across 4 independent experiments (4 mice) in each group (see the graph in new Fig. 5c). Once again, we observed a decrease in the JON/A staining per platelet in *Mical1*-deficient platelets, compared to WT platelets.

New Figure 5: (c) Integrin αIIbβ3 activation of *Mical1*^{+/+} or *Mical1*^{-/-} platelets under arteriolar shear rate (1,500 s⁻¹) (mean ± SD, N = 600 platelets, from 4 independent experiments, two-tailed unpaired Student's t-test, t = 12.30, df = 1198). Left panels: representative images showing JON/A staining of adherent platelets. Scale bars = 10 μm.

Hence, the impaired platelet adhesion and smaller microaggregates observed in MICAL1-deficient platelets on VWF is consistent with the decrease in JON/A binding (a marker of reduced integrin αIIbβ3 activation). Together, the results from Fig. 5 and 7 suggest that the reduction in GPIb-VWF interactions in the absence of MICAL1 compromises the integrin αIIbβ3 activation, thereby impairing the formation of stable platelet adhesion. This mechanism is in line with the model described in the introduction (p. 3), where VWF-GPIbα interactions under shear stress trigger signal that ultimately leads to integrin αIIbβ3 activation necessary to stable adhesion.

We thus added this notion in the discussion (p. 13):

“The GPIb-IX-V complex is responsible for the initial interaction between platelets and VWF, which helps to slow down platelet and thus promote more robust binding to the lesion site through integrin activation⁷¹.”

Finally, we cannot move Fig. 7e to Fig. 5 due to differences in methodology. Specifically, in Fig. 7e, we blocked integrins to study GPIb-dependent rolling velocities. In contrast, in Fig. 5, integrins were not blocked and a stable platelet adhesion, which is integrin-dependent, was observed.

Along the same lines, did they quantify the percentage of rolling platelets on VWF in WT and *Mical1*-deficient platelets?

As requested, we have measured the number of rolling platelets on VWF after perfusing the same number of platelets from both genotypes (similar platelet count between both genotypes, Supplemental Table 2). Our results show a decrease in the number of rolling *Mical1*^{-/-} platelets compared to *Mical1*^{+/+} platelets (see new below and representative Supplementary Video Files 6 and 7).

New Figure 7: (e) Number of rolling platelets per field of *Mical1*^{+/+} and *Mical1*^{-/-} platelets from 3 independent experiments corresponding to different mice (N: *Mical1*^{+/+} = 30, *Mical1*^{-/-} = 30 fields (10 per experiment)), two-tailed unpaired Student's t-test, $t = 3.864$, $df = 58$).

This result is consistent with previous velocity measurements (Fig. 7f) and confirms that MICAL1 promotes VWF/GPIb interactions under shear.

The new data are presented in the new Fig. 7e and in the Results section on p.10:

« The number of rolling platelets from *Mical1*^{-/-} mice was significantly decreased by 50%, compared to control mice (Fig. 7e). Furthermore, *Mical1*^{-/-} platelets exhibited significantly higher translocation velocities ($2.06 \pm 1.04 \mu\text{m/s}$) than *Mical1*^{+/+} platelets ($1.41 \pm 0.89 \mu\text{m/s}$), indicating reduced VWF-GPIb α interactions (Fig. 7f). »

8) I am not convinced that the lipid raft part is backed up sufficiently by experimental data.

a) The literature cited to introduce the concept is relatively outdated, and it is unclear whether modern techniques would still support these claims.

These are important questions that we have addressed below, along with the following sub-points. As described in detail below: (1) we have updated the literature and cited recent papers that use modern approaches; and (2) we added new experimental data that support our conclusion that MICAL1 and F-actin depolymerization promote the translocation of GPIb to lipid rafts.

(1) New references and advances in GPIb/lipid rafts

There is increasing evidence that lipid rafts are dynamic lipid domains within the plasma membrane that play a critical role in the signaling function of numerous receptors (new references #78, 79). Furthermore, it is now well established that F-actin plays an important role in organizing these lipid domains and thereby controls signaling from receptors in these domains (new references # 57-59).

In the case of platelets, different labs using various experimental approaches have reported the functional importance of the translocation/association of the GPIb complex with lipid rafts.

Indeed, the interaction between VWF and GPIb α in platelets was first demonstrated to depend on translocation to cholesterol-rich membranes (lipid rafts), using sucrose density gradient fractionation and Cholera Toxin B as a marker of lipid rafts (reference #13 and new reference #15). Furthermore, the GPIb α within lipid rafts and its association with the actin cytoskeleton have been described (reference #14).

Subsequently, cutting-edge experimental approaches based on Förster resonance energy transfer (FRET) and time-gated fluorescence lifetime imaging microscopy confirmed that GPIb α translocates to lipid rafts (identified with CTxB) during platelets perfusion at a shear rate of $1,600 \text{ s}^{-1}$ on VWF (reference #16).

Next, subcellular fractionation demonstrated that the protein PKA-I localizes to platelet membrane lipid rafts to regulate the phosphorylation of the GPIb β and plays a key role in inhibiting VWF-stimulated platelet adhesion under flow (new reference #11).

Very recently, in a paper published in *Blood*, immunofluorescence revealed an abnormal membrane distribution of GPIb α in cold-stored platelets, which was related to a defect in lipid raft microdomain formation (new reference #12). These new references have now been cited in the introduction (p.3). Finally, recent proteomic data show that the interplay between lipid rafts and cytoskeletal regulators is modified upon platelet activation (new reference #77). From a broader perspective, agent-based modelling approaches have further demonstrated the functional role of lipid raft domains in modulating platelet receptor behavior (GPVI receptor) (new reference #79). This revealed that lipid raft domains significantly influence receptor diffusivity. These two new references have now been cited in the discussion (p.14).

Together, these findings strongly support the view that lipid rafts are essential functional platforms in platelets that regulate signaling dynamics, particularly for GPIb.

(2) New experimental data supporting a role for MICAL1-mediated F-actin depolymerization in GPIb translocation to lipid rafts

To study the interplay between GPIb and lipid rafts, we used spinning disk confocal microscopy and fluorescent Cholera Toxin B (CTxB), rather than the older, more global approaches of lipid fractionation and resistance to detergent. CTxB is one of the most widely used marker for lipid rafts. To clarify this point, we have now cited a recent review regarding the labelling of lipid rafts on p.11 and p.17 (new reference #60). We also changed the labeling of Fig. 8 from “Lipid rafts” to “CTxB”, to better describe the tool that we used to visualize lipid rafts.

In this manuscript, we investigated how the actin dynamics, particularly the disassembly of F-actin by MICAL1, influences the association of GPIb with lipid raft domains. Using spinning disk confocal microscopy and quantification at the level of individual platelets, we found that MICAL1 promotes the partitioning of GPIb into lipid rafts. This provides a mechanistic link between actin remodeling, receptor compartmentalization and the adhesion defects observed upon MICAL1 depletion.

To further show the validity of our microscopy approach, we conducted new experiments in which WT platelets were treated with jasplakinolide prior to perfusion over a VWF matrix. Interestingly, Jasp treatment resulted in a significant reduction in GPIb colocalization with lipid rafts labeled with CTxB, mirroring the phenotype observed in MICAL1 KO platelets.

The new Jasp data have been included in the Fig. 8a and the Results section (p. 11).

These results directly indicate that proper GPIb incorporation into lipid rafts requires actin dynamics, specifically actin depolymerization. Combined with the reduction in GPIb/CTxB colocalization in MICAL1^{-/-} platelets (Fig. 8b) and the rescue of this phenotype through a LatA treatment (Fig. 8c), the new Jasp data in Fig. 8a strengthen our conclusion that MICAL1 promotes GPIb translocation to lipid rafts by depolymerizing F-actin.

New Figure 8a: Colocalization of Cholera Toxin (Red) and GPIb α (Cyan) in *Mical1*^{+/+} platelets treated with either Jasp (2 μ M) or DMSO as control after flow assays at 1,500s⁻¹ from 3 independent experiments corresponding to different mice. Left panels: representative images (scale bars: 5 μ m) and zoom (scale

bars: 2 μm). Manders' overlap coefficient (mean \pm SD, N = 19 fields from 3 independent experiments, two-tailed unpaired Student's t-test, t = 5.563, df = 36).

Collectively, our findings support a model in which MICAL1 and the depolymerization of F-actin contribute to the flow-dependent association of GPIb to lipid rafts, thereby facilitating efficient platelet adhesion.

b) The resolution of the images in Fig. 7i & 7j does not seem sufficient to assess co-localization. The staining appears largely homogeneous, making it difficult to draw firm conclusions.

To assess changes in the association of GPIb with lipid rafts, we used:

- fluorescent CTxB, a widely used, specific lipid raft marker for microscopy (new reference #60)
- a sensitive camera (sCMOS Teledyne Photometrics) with a large dynamic range (16-bit)
- a spinning-disk confocal microscope with a 100x objective having a high numerical aperture (high resolution) of NA= 1.4.

To our knowledge, this is a state-of-the-art approach. For instance, an expert lab in lipid microdomains (Lamaze Lab) recently used similar microscopy and methodology to assess changes in the association of receptors with lipid rafts in immune cells (Belabed et al. Nature Immunology 2025 PMID: 39838105 ; DOI: 10.1038/s41590-024-02065-8).

Of note, the Mander's coefficient can quantify relevant, local differences, by analyzing the overlap between the GPIb and CTxB signals at the level of each pixel.

Notably, we observed reduced GPIb/CTxB overlap in *Mical1*^{-/-} platelets (Fig. 8b) and also in *Mical*^{+/+} platelets when F-actin was stabilized by Jasp (see new Fig. 8a and reply above). Consistent with the idea that MICAL1 promotes GPIb/CTxB colocalization via F-actin depolymerization, the decreased GPIb/CTxB overlap in *Mical1*^{-/-} platelets was rescued by LatA treatment (Fig. 8c). This shows that the microscopy-based methodology that we used is sensitive enough to reveal both increased and decreased of GPIb association to specific lipid domains resulting from drug or genetic perturbations.

Altogether, the previous data, along with the new data, support our conclusion that MICAL1 disassembly of F-actin promotes the translocation of GPIb to lipid domains labeled by CTxB in platelets.

c) The observed difference in the Manders' coefficient could be attributed to differences in tether formation frequency rather than true changes in lipid raft localization (as tethers appear cholera toxin-negative). What happens if only the platelet body is considered?

Although lipid raft staining is more challenging to detect in tethers due to their small diameter, the staining is present and can be quantified (see pictures below).

Figure not included: Representative images from *Mical1*^{+/+} and *Mical1*^{-/-} platelets showing raft staining by cholera toxin subunit B-28 AF555 (CT-B). White arrows show tethers. Note that the contrast has been enhanced to better show the staining in tethers.

We have now specifically measured colocalization in either the platelet body or the tethers. We observed a decrease in colocalization in both regions of MICAL1^{-/-} platelets compared to MICAL1^{+/+}. Therefore, we have kept our analysis of entire platelets in the main Fig. 8, and we have added the analysis performed separately in the platelet body and in the tethers in the new Supplementary Fig. 9.

New Supplementary Figure 9: MICAL1 modulates GPIb α and lipid raft colocalization in the platelet body and tether

Colocalization of lipid rafts and GPIb α in *Mical1*^{+/+} and *Mical1*^{-/-} platelets after flow assays at 1,500s⁻¹ from 3 independent experiments corresponding to different mice. (a) Manders' overlap coefficient for *Mical1*^{+/+} vs *Mical1*^{-/-} platelets within the platelet body (mean \pm SD, N = 16 fields from 3 independent experiments for both genotypes, two-tailed unpaired Student's t-test, t = 3.044, df = 30) (b) Manders' overlap coefficient for *Mical1*^{+/+} vs *Mical1*^{-/-} platelets in the platelet tether (mean \pm SD, N = 16 fields from 3 independent experiments for both genotypes, two-tailed unpaired Student's t-test, t = 3.164, df = 30)

We added this information in the Results section as follow (p.11):

“This reduction in colocalization occurred in both the platelet body and the tethers (Supplementary Fig. 9).”

d) Cholesterol depletion with methyl- β -cyclodextrin (Supplemental Figure 8) affects the entire plasma membrane, not just lipid rafts. The claim that “lipid raft disruption inhibits platelet adhesion on VWF” seems overstated.

We apologize for this overstatement and have changed the sentence to “cholesterol depletion inhibits platelet adhesion to VWF” in Supplemental Fig. 8. We have also modified the text in the Results section (p. 11):

“Consistently, cholesterol depletion induced by methyl- β -cyclodextrin treatment completely disrupted platelet adhesion and stability on VWF (Supplementary Fig. 8), as previously described^{13,16}. It is now well established that F-actin plays a critical role in organizing these lipid domains and controls signaling from receptors in these domains⁵⁷⁻⁵⁹. We thus hypothesized that the excess of F-actin associated with GPIb α in MICAL1-deficient platelets might modify GPIb α association into lipid rafts enriched in cholesterol.”

Minor comments:

1) Western blot images should indicate molecular weights.

We have now indicated the molecular weights in all the Western blots.

2) Clarification for Figure 1a: What does “platelet adhesion in %” refer to? Is it the fraction of total platelets adhering or the surface coverage? If the latter, the reported percentages seem rather low.

« Platelet adhesion % » refers to the % of surface covered by platelets. In the relevant figures, we have now modified the legend with: Platelet adhesion (% of surface coverage). The results are in accordance with the fact that we used recombinant mouse VWF rather than VWF purified from human plasma, as we did for human platelets. Similar results were obtained by different experimenters in the lab.

3) Figure 3d: Were the results obtained after 3 or 6 minutes?

In the first version, Fig. 3d corresponded to flow cytometry experiments in which there was no kinetics. Our data were obtained by flow cytometry and were recorded 20 min after incubation with antibodies, as described in the Material and Methods section. We believe there is an error in the figure cited by the reviewer. If the reviewer is referring to Fig. 2c-d (immunoprecipitation of GPIb), the results were obtained 2 minutes after VWF/Ristocetin activation. We have now specified this timing in the legend of Fig. 2c-d.

4) Rephrase page 6, line 14: Fig. 4a refers to ex vivo thrombus formation. The current phrasing “in Mical1^{-/-} mice” is misleading.

On p. 7 we changed the sentence to:

“Remarkably, platelet thrombus formation was significantly altered in the absence of MICAL1 compared to controls. In particular, individual thrombi were larger in the absence of MICAL1, as evidenced by increased thrombus volume (Fig. 4a).”

5) Piezo1 as a potential mechanosensitive factor: The authors performed a comprehensive analysis of actin regulators (Supplemental Fig. 5), but did they assess whether Mical1 affects Piezo1 expression? This could provide an additional link to the observed defects in mechanotransduction.

As requested, we performed a Western-blot using a PIEZO1 antibody. No difference was observed between WT and KO platelets, assessed in 5 WT and 5 KO mice.

Expression of PIEZO-1 by Western-blot in WT and KO MICAL1 platelets (N: WT=5, KO=5).

This new result is now included in Supplementary Fig. 5d-e.

New Supplementary Figure 5: *Mical1* deficiency does not affect F-actin polymerization in static conditions and does not impact the expression of main actin regulators and other proteins related to platelets or *Mical1*

(d) Representative images of western blots of lysates from *Mical1*^{+/+} and *Mical1*^{-/-} platelets.

(e) Relative expression of proteins in *Mical1*^{-/-} compared to *Mical1*^{+/+} set to 1 (red dashed line). (N: β-actin = 3, γ-actin = 6, Myh9 = 5, MyI9 = 3, α-tubulin = 3, β-tubulin = 5, α-actinin = 3, Cofilin = 3, Zyxine = 3, ARP2C = 3, Paxillin = 2, FLNA = 6, IQGAP1 = 6, WASp = 4, RhoA = 8, CDC42 = 3, RAC1 = 3, Talin = 5, Kindlin-3 = 3, DYNC1/2 = 4, MAPRE2 = 4, MTCO2 = 4, αIIb = 3, β3 = 5, MPL = 3, PIEZO1 = 5, Fg = 5, Rab35 = 3, Rab8a = 4, PLXB2 = 3, Vimentin = 6, ALIX = 3, MKLP1 = 4, EHD3 = 5, CAMK2G = 3, with 14-3-3z as loading control).

ns: not significant. N = number of independent experiments from different mice.

6) Please indicate flow direction in Figure 6.

We have now indicated the flow direction in Fig. 6

Reviewer #2 (Remarks to the Author):

It has been well established that actin dynamics in mammalian cells is controlled by monooxygenase activity of MICAL enzymes that oxidize specific actin methionine residues which results in actin filament disassembly. In this study, Mical1 has been linked to actin filament disassembly in platelets mediated by the VWF-GP1b interaction under high fluid shear. The studies have been well designed, the data is robust and the results support the conclusions.

Comment

Platelet interactions in the developing thrombus are dynamic where they can undergo several cycles of adherence, rolling and/or detachment. The studies are presented as a linear sequence of irreversible events, whereas all steps are potentially (likely) reversible. Notably, actin oxidation by MICALs has been shown to be reversed by methionine sulfoxide reductases (MSR) in other cells. It is plausible that actin filament disassembly in platelets is a balance between MICAL and MSR activities at any stage.

We fully agree with the reviewer. The regulation of F-actin in the developing thrombus is dynamic, and the balance between actin oxidation by the oxidase MICAL1 and actin reduction by the methionine sulfoxide reductases (possibly MSB2, as we previously reported during cytokinesis, reference #38) is likely important. We have added the following sentences to the discussion section to emphasize that this is a dynamic and reversible process (p. 14):

“Of note, the changes in F-actin disassembly during platelet activation are reversible⁸⁰ and likely depend on a dynamic balance between MICAL1-mediated oxidation and Methionine Reductase (MSRB)-mediated reduction of actin, as observed during bristle development in drosophila or cytokinesis in human cells^{81,31,38,25}.”

Reviewer #3 (Remarks to the Author):

The manuscript by Solarz et al. studied the role of Mical1, an enzyme involved in the disassembly of F-actin, and subsequent platelet adhesion and stability in response to high shear flow. The authors examine the role of actin polymerization and depolymerization with actin drugs and a Mical1 knockout mice used for in vivo and in vitro experiments. In the presence of shear flow, they found that F-actin disassembly associated with the GPIb-IX-V complex promotes platelet adhesion and stability, while F-actin assembly has the opposite effect. Additionally, the data suggests that Mical1 enhanced platelet adhesion, thrombus stability, and occlusion at the lesion site. Without Mical1, embolization was more likely to occur. Additionally, Mical1^{-/-} platelets were shown to have morphological changes, including elongated membrane tethers and increased F-actin levels, and decreased localization of GPIb α in lipid rafts. The authors concluded that Mical1 regulates shear-dependent actin reorganization within the GPIb-IX-V complex, which plays a role in platelet adhesion and stability. While a strong manuscript, this reviewer has concerns about overstatements in the title and text, additional conditions to support the author's conclusions, and characterization of the Mical1 KO.

MAJOR COMMENTS

1. Title. The title of the manuscript is “Mechano-dependent VWF-GPIb α interaction in platelets requires F-actin disassembly by the oxidoreductase Mical1.” This reviewer is convinced that Mical1 is involved but sees “require” as an overstatement given that Mical1^{-/-} platelets still bind to VWF under shear. This reviewer suggests the authors modify the title and soften the “require”.

As requested, we have changed the title to:

“F-actin disassembly by the oxidoreductase MICAL1 promotes mechano-dependent VWF-GPIb α interaction in platelets”

2. LatA and Jasp characterization.

a. From the results in Fig. 2b, the authors claim that “the redistribution of Mical1 and GPIb α from the soluble to the insoluble cytoskeletal fraction was abolished upon F-actin depolymerization by LatA”. However, this reviewer sees a distinct band in the blot image for Mical1 +LatA at 2 mins, which does not seem to be represented in the graphs or support the “abolished” claim. Additionally, error bars are missing (or perhaps are too small to see) for several of the data points in Fig. 2b. To make the conclusion more convincing, the authors could consider a combination of the following approaches (1) include a table with the raw western blot values, (2) plot the +LatA lines on a separate graph with a smaller scale to show the true signal and error bars, (3) demonstrate the effect on the Triton X-100 insoluble fraction when F-actin is stabilized with +Jasp.

As suggested, we have now (1) included a table with the raw Western blot values in the Supplementary Raw Data Excel file (the error bars were indeed sometimes too small to be seen) and (2) plotted the +LatA lines on a separate graph with a different scale to show the true signal and error bars, as shown in the new Fig. 2b below.

New Figure 2b: (b) Lysates of human platelets treated or not with LatA (15 μ M) at 0 s (resting platelets), after 30 s or 2 min of activation (VWF 10 mg/mL/Risto 0.4 mg/mL) were separated into Triton X-100 soluble and insoluble fractions and blotted. Quantification of insoluble MICAL1 (mean \pm SD, N = 4 independent experiments from 4 different donors, two-way ANOVA with Šídák post hoc test, $F(2, 18) = 27.06$, 2 min: $p < 0.001$), GPIIb/IIIa (mean \pm SD, N = 3 independent experiments from 4 different donors, two-way ANOVA, $F(2, 12) = 49.40$, 2 min: $p < 0.001$) and β -actin (mean \pm SD, N = 4 independent experiments from 4 different donors, two-way ANOVA with Šídák post hoc test, $F(2, 18) = 2.424$, 2 min: $p < 0.001$). Left panel: representative western blots. Right panels: MICAL1, GPIIb/IIIa, β -actin quantification from western blots.

We agree with the reviewer that “abolished” was an overstatement and have corrected the main text as follows (p. 6):

“Furthermore, the redistribution of MICAL1 and GPIIb/IIIa from the soluble to the insoluble cytoskeletal fraction was strongly diminished upon F-actin depolymerization by LatA (Fig. 2b).”

As requested, experiments were also conducted by treating platelets with Jasplakinolide. In resting platelets, actin was found more abundant in the Triton-insoluble fraction, as expected, while no accumulation of MICAL1 was observed. In contrast, both actin and MICAL1 levels were increased following activation with VWF/Ristocetin. These results suggest that the accumulation of F-actin induced by Jasplakinolide is not sufficient to recruit MICAL1 to the insoluble fraction. Furthermore, this supports the idea that additional signals depending on platelet activation are required to mediate the association of MICAL1 with actin. However, we acknowledge that this observation may be beyond the main scope of the study. We have therefore combined point 1 and point 2 in the revised Figure and did not include point 3 for conciseness. However, we can include this additional data in the Supplementary section if requested by the reviewer.

Figure not included: MICAL1 and β -actin presence in the Triton X100-insoluble fraction before and after activation with VWF/Ristocetin following LatA or Jasp treatment. (mean \pm SD, N = 3 independent experiments, two-way ANOVA with Šídák post hoc test, MICAL1: $F = 49.5$, $df = 12$; β -actin: $F = 133.7$, $df = 12$).

b. In supplemental Figure 1b, the authors show that LatA results in destabilization of F-actin and that Jasp results in stabilization of actin. However, this conclusion and effect size is based on N = 1 with no statistics. An estimate of the effect size at the chosen concentrations is important to the conclusions of this work; N = 1 with no statistics is insufficient.

As requested, we added three more experiments to confirm the efficacy of the drugs LatA and Jasp in platelets (new Supplementary Fig. 1c below).

New Supplementary Figure 1: (c) Mouse platelets were treated with either Latrunculin-A (LatA; 500 nM), Jasplakinolide (Jasp; 1 μ M), or DMSO as a control, lysed and β -actin was analyzed in both the Triton-X100 insoluble and soluble fractions to assess the effects of the drugs on actin solubility, reflecting polymerization or depolymerization status. Of note, F-actin is incorporated into the Triton-X100 insoluble fraction. (mean \pm SD, N = 4, one-way ANOVA with Tukey post hoc test, F = 12.33, df = 11)

c. The conclusion that Mical1 disassembles F-actin and enables GPIb translocation into lipid rafts is critical to this paper (Figure 7). This conclusion would be strengthened if both Mical1^{+/+} and Mical1^{-/-} were treated with LatA and Jasp and GPIb/lipid raft was measured. The condition that is especially important is Mical1^{+/+} platelets treated with Jasp.

We would like to thank the reviewer for these suggestions to strengthen our conclusions.

As requested, we conducted new experiments in which WT platelets were treated with jasplakinolide prior to perfusion over a VWF matrix. Interestingly, Jasp treatment resulted in a significant reduction in GPIb colocalization with lipid rafts in WT platelets, mirroring the phenotype observed in MICAL1 KO platelets.

These new results are now included in Fig. 8a, as shown below:

New Figure 8a: Colocalization of Cholera Toxin (Red) and GPIb α (Cyan) in Mical1^{+/+} platelets treated with either Jasp (2 μ M) or DMSO as control after flow assays at 1,500 s⁻¹ from 3 independent experiments corresponding to different mice. Left panels: representative images (scale bars: 5 μ m) and zoom (scale bars: 2 μ m). Manders' overlap coefficient (mean \pm SD, N = 19 fields from 3 independent experiments, two-tailed unpaired Student's t-test, t = 5.563, df = 36).

These results directly indicate that proper GPIb incorporation into lipid rafts requires actin dynamics, specifically actin depolymerization. Combined with the reduction in GPIb/CTxB colocalization in MICAL1^{-/-} platelets (Fig. 8b) and the rescue of this phenotype through a LatA treatment (Fig. 8c), the new Jasp data in Fig. 8a strengthen our conclusion that MICAL1 promotes GPIB translocation to lipid rafts by depolymerizing F-actin.

As mentioned by the reviewer, the condition that is especially important is Mical1^{+/+} platelets treated with Jasp (new Fig. 8a). We have not performed Jasp treatment in Mical1^{-/-} platelets because our aim was to understand the general role of actin using wild-type mice (Fig. 1). We also wanted to avoid inducing excessive actin polymerization by combining Jasp and MICAL1 depletion and, as such, obtaining an extreme phenotype. However, as suggested, we did a rescue assay by treating Mical1^{-/-} platelets with low doses of LatA (Fig. 8c), as mentioned above.

3. Mical1^{-/-} characterization.

a. In Figure 3a, it appears that the Mical1^{-/-} has substantially more alpha-tubulin. In supplemental figure 5, it appears that there is no difference in alpha-tubulin. This reviewer suggests that the authors provide some context, explanation, and/or additional evaluation.

This apparent discrepancy can be explained.

Western blot experiments using 14-3-3 as a loading control indeed revealed comparable alpha-tubulin levels between WT and MICAL1 KO platelets (see Supplementary Fig. 5d and quantifications in Supplementary Fig. 5e).

In Fig. 3a, the aim was to show that there was no detectable MICAL1 protein in Mical1^{-/-} platelets and we loaded a higher amount of lysates from the KO compared to the WT samples, to make sure that potential small amounts of MICAL1 would not be missed. This explains the more intense signal for the loading control in the KO lanes in this WES assay (capillary electrophoresis). This has now been clarified in the Fig. 3a legend.

b. The authors find that mean platelet volume is not significantly different with and without Mical1 (Supplementary Table 2), but that platelet surface area is significantly higher in the absence of Mical1 ($p < 0.001$, Figure 3b). This is a key point that should be further investigated. Either (1) the platelet shape is different in the absence of Mical1, such that it is possible to have a different surface area but the same volume OR (2) the volume per platelet is different, but the difference is not identifiable when comparing average platelet volume per mouse instead of examining platelet volume on a per platelet basis (similar to what is done in Figure 3b). Based on the image in Figure 3c, the higher surface area of Mical1^{-/-} platelets, and the trending (though not significantly lower) MPV of Mical1^{-/-} platelets, this reviewer suspects that platelets lacking Mical1 have a significantly more oblong shape.

We thank this reviewer for this insightful and important comment. As suggested, we conducted additional analyses to further investigate the potential differences in platelet morphology between control and Mical1^{-/-} mice.

We have now measured both the major and minor diameters of individual platelets from the imaging dataset shown in Fig. 3. These additional measurements enabled us to characterize the shape and size of platelets more accurately at the single-cell level.

Our results indicate that the major and minor diameters are both significantly increased in Mical1^{-/-} platelets compared to controls, suggesting that these platelets are indeed slightly larger overall. This is consistent with the previously observed increase in surface area. Importantly, the ratio between the major and minor axes did not differ significantly, indicating that the platelet shape remains relatively

round overall and that the increase in surface area reflects an isotropic increase in size rather than a shift towards a more elongated or oblong morphology. The observation that the volume is not significantly increased is likely due to the magnitude of the size increase and limitations in resolution of the automated measurements.

We have included these new results in new Fig. 3c-e.

New Figure 3 (c-e) Platelet ultrastructure analyzed by transmission electron microscopy from 2 independent experiments from a pool of three mice in each experiment: (c) Measurement of the major axis of *Mical1*^{+/+} and *Mical1*^{-/-} platelets. Violin plots with the median represented by a central line and the interquartile range (25th-75th percentiles) indicated by the upper and lower lines. (N = 200 platelets per genotype; two-tailed unpaired Student's t-test, $t = 2.706$, $df = 398$). (d) Measurement of the major axis of *Mical1*^{+/+} and *Mical1*^{-/-} platelets. Violin plots with the median represented by a central line and the interquartile range (25th-75th percentiles) indicated by the upper and lower lines. (N = 200 platelets per genotype; two-tailed unpaired Student's t-test, $t = 2.699$, $df = 398$). (e) Right panel: representative diagram of the major and minor axis of *Mical1*^{+/+} and *Mical1*^{-/-} platelets.

These new data support the conclusion that the absence of MICAL1 results in slightly larger platelets, without drastic changes in shape. This analysis is described in the revised manuscript (p. 6) as follows:

“Of note, transmission electron microscopy (TEM) analysis revealed a slight increase in platelet size in only a subset of the population (12% of platelets) (Fig. 3b-f). To further characterize platelet morphology, we quantified both the major and minor diameters of individual platelets and found that both parameters were significantly increased in *Mical1*^{-/-} platelets compared to controls, indicating a moderate but consistent increase in platelet size. However, the ratio between the major and minor axes remained unchanged, suggesting that platelet shape was preserved and that platelets in *Mical1*^{-/-} mice are slightly larger overall (Fig. 3c-e). Consistent with this, the mean platelet volume (MPV) measured by hematology analyzer was not significantly different in *Mical1*^{-/-} platelets (Supplementary Table 2), likely due to the magnitude of the size increase and limitations in resolution of the automated measurements (Supplementary Table 2). Moreover, the surface expression of major platelet receptors, including GPIIb α , was normal (Fig. 3g). Other parameters such as platelet clearance and the number of megakaryocytes in the bone marrow were also normal (Supplementary Fig. 2). We conclude that absence of MICAL1 leads to slightly larger platelets without significant alteration in shape, and that platelet count, clearance and receptors expression remain normal in *Mical1*^{-/-} mice.”

c. In supplementary Figure 2c, the authors find that *Mical1* deficiency does not affect number of MKs in the bone marrow based on N = 3 mice. This reviewer is not convinced of this conclusion. Estimating from the graph, the number of MK per square mm with *Mical1* is 17 and the number without is 12 (estimated 1.4-fold difference). This is a similar effect size to what is observed in carotid artery occlusion time (Figure 3f), yet is ns due to the lower quantity of mice characterized. The authors should provide a compelling rationale regarding why N = 3 was sufficient here (but not elsewhere throughout the paper) or should conduct additional experiments to better characterize this apparent 1.4-fold difference.

First, we would like to clarify the methodology. Specifically, the number of megakaryocytes (MK) was calculated using an *entire femur section* from each mouse, rather than just a single view. To account for this fact, all views measured in both genotypes are now displayed in the revised Supplementary Figure 2c. As previously shown, when a total of 10 views per group (from 3 mice) were included, no significant difference was observed. Although a trend appears in the graph, the p-value (0.1280) indicates that the difference is not statistically significant.

The reason we did not include more animals in this group was due to ethical considerations based on our biological observations and the principles of the 3Rs. Indeed, regarding the platelet count, we had found no differences between WT and KO mice when large groups of 24-34 mice were measured, indicating the absence of clinical manifestations such as thrombocytopenia or thrombocytosis (see Supplemental Table 2). As platelet count reflects the balance between production and elimination, the lack of significant differences suggests that any potential effect on the bone marrow would be minimal and biologically irrelevant. Furthermore, we measured platelet lifespan and β -galactose exposure, which is involved in platelet clearance. Again, we found no variations between WT and KO mice (see Supplementary Fig. 2a, b).

Finally, since the spleen is a haematopoietic organ in mice that can compensate for bone marrow deficiency, we had measured the spleen mass to assess the presence or absence of splenomegaly. These results, which were not included in the manuscript, showed no significant difference between the WT and KO groups, suggesting an absence of bone marrow defects compensated by the spleen.

Figure not included: The spleen weight is normal in absence of MICAL1 in mice. The spleen weight is calculated as a ratio with the body weight of each mice. N=17 WT mice, 16 KO mice. Unpaired T test $p=0.7397$.

Together, these results support the conclusion that there are no alterations in MK maturation or platelet elimination in KO mice, and that their platelet count remains within the normal range.

4. Overstatement in main text.

a. In a subtitle of the main text and figure title of Fig. 3, the authors summarize their findings as “Mical1 is required for hemostasis and thrombosis in vivo”. This reviewer sees this as an overstatement not sufficiently supported by the data. However, this reviewer would agree that the data indicates Mical1 contributes to hemostasis and thrombosis in high shear conditions. For example, hemostasis is reached in Figure 3e, albeit with a longer bleeding time, even in the absence of Mical1. Also, Fig. 3h does not seem to support this claim without indicating ‘high shear conditions’. Further, this reviewer suggests adding more as to why there are differences in the effects of Mical1 in the carotid artery, mesenteric arteriole, and mesenteric venule.

We agree with the reviewer, and have changed the title of Fig. 3 to:
“MICAL1 contributes to hemostasis and thrombosis under high shear.”

As suggested, we have added the shear rates for the conditions shown in Figure 3i-k *in vivo*, taken from the literature. According to the literature, the shear rates for the intact carotid artery and intact mesenteric arteries are reported as 950 s^{-1} (new reference #53) and $1,400 \text{ s}^{-1}$ (new reference #54), respectively. For the intact mesenteric vein, new reference #52 indicates a shear rate range between $80\text{-}100 \text{ s}^{-1}$.

The following sentence has been added to the results on p. 7:

« We next assessed thrombus formation following FeCl₃-induced injury in different vessels with different shear rates: mesenteric venules (low shear rates; 100 s⁻¹)⁵², carotid artery (moderate shear rates: 950 s⁻¹)⁵³ and mesenteric arterioles (high shear rates: 1,400 s⁻¹)⁵⁴. »

b. In Supplementary Fig. 3a, b this reviewer does not agree that the data fully support the claim that “agonist-induced platelet aggregation was significantly enhanced in *Mical1*^{-/-} platelets due to increased release of dense granules”. For several concentrations of agonists (Supplementary Fig. 3a), there is no significance between *Mical1*^{+/+} and *Mical1*^{-/-} and ADP shows no significance between aggregation for the two conditions. Only Supplementary Fig. 3b seems to show enhanced dense granule release via ATP concentration in response to thrombin.

We agree with the reviewer and have revised the statement accordingly.

We acknowledge that ADP, which is considered as a weak agonist, does not induce secretion in these conditions (i.e. washed platelets), unlike other tested agonists. The absence of a difference in aggregation induced by ADP, together with the increased aggregation induced by stronger agonists (Thrombin, Cvx, PAR4-AP, which induce the secretion necessary to promote aggregation), in *Mical1*^{-/-} compared to WT mice supports the hypothesis that this increased aggregation phenotype is related to increased secretion rather than another mechanism (such as a direct inside-out signaling, which is necessary for αIIbβ3 integrin activation). If such a mechanism were at stake, the aggregation induced by ADP-induced would have been different between in *Mical1*^{-/-} vs. WT mice.

To confirm the role of MICAL1 in secretion in the aggregation phenotype, we have now performed aggregation assays in the presence of a high dose of apyrase to remove both ATP and ADP, as recommended by reviewer #1.

In this condition, we restored normal aggregation in MICAL1 knockout platelets to levels comparable to those of WT platelets (see Supplementary Fig. 3c, d). Furthermore, treating platelets with apyrase under flow showed that the increased aggregation of MICAL1^{-/-} platelets was also dependent on secretion in flow conditions (see Supplementary Fig. 3a). These data indicate that the enhanced aggregation observed in the absence of MICAL1 is a result of increased secretion.

We have changed the sentence in the results section (p. 7), and the revised Supplement Figure 3c-d now includes these data.:

“Similarly, platelet aggregation and ATP secretion were significantly enhanced in *Mical1*^{-/-} platelets induced by Thrombin, Cvx, PAR4-AP agonists leading to secretion (strong agonists) but not by ADP (weak agonist) (Supplementary Fig. 3b). In line with the observed ex vivo thrombus formation, treatment with apyrase normalized the aggregation of *Mical1*^{-/-} platelets to the level observed *Mical1*^{+/+} platelets. These results indicate that the aggregation enhancement is primarily due to increased release of dense granules (Supplementary Fig. 3c, d).”

Figures included in new supplemental Figure 3: (a) Rhodamine 6G stained *Mical1*^{+/+} or *Mical1*^{-/-} platelets in whole blood treated with 5 U/mL of apyrase were perfused on type I collagen matrix. Thrombus size at a shear rate of 1,500 s⁻¹ (mean fluorescence intensity (MFI) ± SD, N = 5; two-tailed

paired Student's t-test, $t = 0.04228$, $df = 8$). Left panel: representative images. (d) Aggregation of washed *Mical1*^{+/+} and *Mical1*^{-/-} platelets activated with thrombin 600 nM after incubation with 2 U/mL of apyrase or not (mean \pm SD, in absence of apyrase $N = 3$, in presence of apyrase $N = 6$, multiple unpaired Student t-test, t : without apyrase (-) = 0.3269, with apyrase (+) = 0.2774, $df = 14$). Left panel: representative aggregation traces.

c. In Fig. 5f, 5g, the authors conclude that “Mical1 deficiency induces excessive F-actin polymerization”; however, this reviewer sees this conclusion as a stretch for the data shown. The figure only shows the defects in the adhesion and stability of Mical1^{-/-} platelets. This statement is better supported by the results shown in Fig. 6.

We would like to thank the reviewer for this comment and for giving us the opportunity to clarify our interpretation.

While we agree that the F-actin measurements shown in Figure 6 provide more direct support for the conclusion regarding excessive F-actin polymerization in *Mical1*-deficient platelets, we would like to emphasize that Figure 5d already showed an increased F-actin signal in adherent *Mical1*^{-/-} platelets under flow conditions.

In addition, we previously did rescue experiments using drugs to functionally show that the increased levels of F-actin in MICAL1 KO platelets are likely responsible for the observed phenotype. To achieve this, we used low doses of Latrunculin-A which are insufficient to affect adhesion in WT cells, but which rescued adhesion in MICAL1 KO platelets (Fig. 5f-g).

Collectively, the results presented in Fig. 5f-g support the notion that the increased F-actin content observed in Figure 5d contributes to the adhesion and stability defects.

Nonetheless, we have added “see also below” to inform the reader that this conclusion is supported by other data in Fig. 6 and 7. The sentence mentioned by the reviewer has been revised as follows (p. 9):

“Thus, MICAL1 deficiency induces excessive F-actin polymerization (see also below), which is responsible for the defects in adhesion and stability of *Mical1*^{-/-} platelets”.

MINOR COMMENTS

1. Abstract. In the abstract, the authors state, “but whether actin dynamics is important for platelet function under hemodynamic, high shear conditions, is largely unknown”. This reviewer sees this statement on the gap in the field as an overstatement. Actin dynamics have been characterized in many conditions under shear such as platelet shape change and spreading, granule secretion, clot retraction and stability, mechanosensing, and force transmission. The authors should rephrase this statement to more specifically state the gap this manuscript fills.

We agree with the reviewer and have indicated in the revised version of the abstract that our aim was to investigate the role of actin dynamics in GPIIb function (changes are underlined):

“Mechano-dependent interactions are essential for thrombus formation and bleeding arrest, as they allow stable platelet adhesion to damaged vessel walls. The interaction between von Willebrand factor (VWF) exposed on wounded vessels and the platelet receptor GPIIb-IX-V is central to this adhesion. The actin cytoskeleton interacts with the cytoplasmic domain of GPIIb, but whether actin dynamics is important for GPIIb function under hemodynamic, high shear conditions, is largely unknown. Here, we found that actin disassembly is a critical determinant for proper VWF-GPIIb interaction mediated by mechanotransduction. Mechanistically, we identified the oxidoreductase MICAL1 as a key enzyme that, specifically under shear, enables the disassembly of F-actin associated with the GPIIb-IX-V complex. This allows the GPIIb-IX-V complex to translocate to lipid rafts, which in turn reinforces VWF-GPIIb interactions. As a result, MICAL1 plays a pivotal role in platelet adhesion to VWF and resistance to cell deformation under shear, and is thus required in both adhesion and thrombus

stability in vivo. These findings establish a role for MICAL1 in the control of shear-dependent actin reorganization within the GPIb-IX-V complex, providing an unexpected connection between actin oxidation and platelet function.”

2. Misleading statements about VWF exerting force. Overall, the introduction is well-written and appropriately cites the existing literature. One constructive comment is that several times, the authors mention that the force between VWF and GPIb is induced by VWF pulling (introduction line 8, line 19). This statement is misleading. VWF is not alive and does not pull (on its own). It is this reviewer’s understanding that tension on the VWF-GPIb bond comes from two main sources: 1) blood flowing over the platelet-VWF bond, pulling on both VWF and platelets and 2) platelet contraction. This reviewer strongly suggests clarifying this point to better frame the paper and to dispel the misunderstanding that VWF pulls on the bond.

The reviewer is right that VWF does not pull by itself. This misleading statement arose from experiments in which optical tweezers were used to exert pulling forces on isolated parts of VWF to demonstrate the mechanosensitive nature of GPIb. As requested, we have made the following changes to the introduction (changes are underlined):

The ability of platelets to adhere to sites of vascular injury is essential for the arrest of bleeding and subsequent vascular repair. Von Willebrand factor (VWF) - a large multimeric protein found in plasma - is a key adhesive protein that initiates platelet-vessel wall interactions under high shear stress¹. Once immobilized on collagen fibers that become accessible at the site of vessel wall injury, the A1 domain of VWF captures rolling platelets from rapidly flowing blood via specific interactions with the platelet glycoprotein (GP) Ib α / β -IX-V complex. Using optical tweezers to exert pulling forces on the isolated A1 domain that binds to the N-terminal part of GPIb α , the unfolding of a juxta-membrane domain—the mechanosensory domain (MSD)—was observed upon continuous pulling force^{2,3}. Although distant from the N-terminal region of GPIb α , unfolding of the MSD enhances the lifetime of the interaction between GPIb α and the A1 domain of vWF⁴. However, this is not sufficient to immobilize platelets at the site of injury, and additional mechanisms are required to withstand the mechanical forces generated by the blood flow. These include the extension of plasma membrane tethers filled with actin filaments that promote platelet capture, which contribute to firm adhesion and spreading⁵⁻⁷. Stable platelet adhesion to VWF also requires the activation of the integrin α IIb β 3, which binds directly to VWF in the RGD peptide motif in the C4 domain^{8,9}. Following adhesion to the site of injury, immobilized platelets aggregate one to another through GPIb-IX-V binding to VWF and α IIb β 3 binding to fibrinogen¹⁰, leading to the formation of a thrombus anchored to the damaged vessel, which occludes the vascular breach and stops bleeding.

VWF engagement with GPIb α under physiological shear stress induces MSD unfolding on platelets and propagates a signal across the plasma membrane to the cytoplasmic domain of GPIb α ². (...).

3. Experimental details. This reviewer suggests

a. Mentioning that mouse blood was used within the main text for experiments in figure 1. This information it is currently only in the figure caption.

We have added the information and we changed the sentence as follows (p. 5):

“To investigate the role of actin dynamics in platelet-VWF interaction under shear, we treated mouse whole blood (...).”

b. Explaining what is meant by “independent experiment.” It is unclear whether an ‘independent experiment’ is a different mouse on a different surface on a different day or whether an ‘independent experiment’ may be the same mouse blood on the same day, but only a different surface. E.g., in the captions of Figure 1a-c and Figure 4a-c.

Where appropriate, we clarified in the figure legends that independent experiments corresponded either to individual mice or to separate human blood donors. It should be noted that the same mouse blood was never used in multiple experiments on the same day.

c. Including many fields of view were captured in each figure. This is done in Figure 5c, but is not done in consistently through the paper (e.g., Figure 1, 4, 5).

We indicated in each relevant figure legend either the mean for each experiment when the analysis was performed by fields (e.g. Fig. 1a), or the distribution within the platelet population when the measurement was performed on each individual platelet (e.g. Fig. 5d).

d. Providing a rationale for revering to shear in terms of s^{-1} and then switching to dynes/cm² in supplementary figure 6

We used shear rate in s^{-1} when referring to blood flow, as this quantifies the rate at which adjacent layers of fluid move with respect to each other, and is the standard unit of measurement in hemodynamics. In contrast, we used shear stress in dynes/cm² in Supplementary Figure 6 to describe the mechanical forces applied directly to cultured cells, as this is the relevant biophysical parameter in *in vitro* experiments.

From a fluid mechanics perspective, shear stress (τ , in dynes/cm²) is related to shear rate ($\dot{\gamma}$, in s^{-1}) by the equation $\tau = \eta \cdot \dot{\gamma}$, where η is the dynamic viscosity of the fluid. Moreover, in simple parallel flow (e.g., between two plates), the shear rate can be expressed as $\dot{\gamma} = 6Q / (wh^2)$, where Q is the volumetric flow rate, w is the channel width, and h is the channel height. In blood, due to its non-Newtonian behavior, shear rate is often used directly to describe flow conditions. In cell culture systems, where the medium behaves as a Newtonian fluid with a known viscosity, shear stress can be calculated and is more biologically relevant as it reflects the actual mechanical force experienced by cells.

e. Identifying a direction of flow on the images in figures 6 and 7.

We have now added arrows.

4. Experimental rationale. This review suggests including

a. The rationale for the 1500 s^{-1} is that this is an arteriolar shear rate. This reviewer suggests adding a rationale about why the higher shear rate of 9000 s^{-1} was selected. The authors may find this publication helpful [10.1182/bloodadvances.2022007550](https://doi.org/10.1182/bloodadvances.2022007550). It would also support the manuscript to provide estimated shear rates for the conditions in Figure 3e-h, either by experimental measurement or estimations from the literature.

In our flow chamber experiment, the rationale for selecting a shear rate of 9,000 s^{-1} was to simulate hemodynamic conditions that can be encountered in regions of high shear stress, such as those found in microvascular and arterial systems *in vivo*.

We chose a shear rate of 9,000 s^{-1} based on a dose-response analysis of platelet stability under increasing shear stress. Our results showed a progressive decline in stability as the shear rate increased, culminating in approximately 50% thrombus stability at 9,000 s^{-1} .

We added this curve to new Supplementary Fig. 1b as follows:

New supplementary Figure 1: Overview of flow experiments and measurement of the effect of Latrunculin-A and Jasplakinolide on actin polymerization

(b) Thrombus stability was evaluated at various shear rates on collagen matrix. Dashed red line indicates 50% of stability.

As indicated in the response to Major point # 4 above, we have now added the estimated shear rates observed *in vivo*. As suggested, we have added the estimated shear rates calculated *in vivo* from the literature. According to the literature, the shear rates for the intact carotid artery and intact mesenteric arteries are reported as 950 s^{-1} (reference #53) and $1,400 \text{ s}^{-1}$ (reference #54), respectively. For the intact mesenteric vein, reference #52 indicates a shear rate range between $80\text{-}100 \text{ s}^{-1}$.

The following sentence has been added to the results on p. 7:

« We next assessed thrombus formation following FeCl_3 -induced injury in different vessels with different shear rates: mesenteric venules (low shear rates; 100 s^{-1})⁵², carotid artery (moderate shear rates: 950 s^{-1})⁵³ and mesenteric arterioles (high shear rates: $1,400 \text{ s}^{-1}$)⁵⁴. »

b. The rationale for how the authors determined the number of biological or technical replicants (n = ?) to include in each study.

A power analysis was performed using G*Power software was performed to ensure that the sample size was sufficient to detect statistically significant differences with a power of 80% and a significance level of 0.05. All mouse procedures, including the statistics and the number of mice necessary for each procedure have been validated and approved by the local ethical committee CEEA26 and the French government under the number APAFIS#25086-2020032312267714.

c. The rationale for why the concentrations of Jasp1 and LatA were selected.

We began with the lower end of the effective dose range to minimize the risk of non-specific effects. In the platelet literature, Jasp and LatA are commonly used at concentrations ranging from 1 to $10 \mu\text{M}$. We opted to test low concentrations of Jasp based on a study reporting that higher doses ($10 \mu\text{M}$) led to shedding of the GPIIb α receptor (Zhou, K. et al. Platelets 33, 381–389, 2022). In our system, we found that $2 \mu\text{M}$ Jasp was sufficient to (1) induce a distinct platelet adhesion phenotype on VWF and (2) promote F-actin accumulation in the Triton-insoluble fraction (Supplemental Figure 1c). A similar rationale was applied to LatA: a low concentration ($0.5 \mu\text{M}$) effectively caused a redistribution of actin to the Triton-soluble fraction, also associated with an adhesion phenotype (Supplemental Figure 1c). It has been described that concentrations as low as $0.5 \mu\text{M}$ can disrupt actin polymerization in mammalian cells (FEBS Lett 1987 Mar 23;213(2):316-8. doi: 10.1016/0014-5793(87)81513-2.) and doses above $1 \mu\text{M}$ may lead to non-specific effects, including alterations in cell morphology and proliferation.

5. Supplementary data. This reviewer commends the authors for providing substantial supplementary information, such as precise product information. This reviewer additionally suggests that the authors add the following to the supplementary data:

a. a supplementary figure with representative raw flow cytometry data used to create Figure 1d, These data have now been included in the Supplementary raw data Excel file.

b. supplementary videos of raw data used to create Figure 1c,

Three videos are now provided:

Supplementary Video File 1: Rolling experiment of mouse platelets treated with DMSO.

Supplementary Video File 2: Rolling experiment of mouse platelets treated with LatA.

Supplementary Video File 3: Rolling experiment of mouse platelets treated with Jasp.

c. supplementary figures of unedited (i.e., no repositioned lanes) blot images for all blots,

We now provide a Supplementary file showing the unedited blots.

d. supplementary videos of raw data used to create Figure 3i,

e. supplementary videos of raw data used to create Figure 4a-c,

f. supplementary videos of raw data used to create Figure 5,

These experiments were not monitored in real time using time-lapse microscopy. Instead, all measurements (occlusion time *in vivo*, thrombus size and stability, % of adhesion in flow) were

performed at the final indicated time (Supplementary Fig. 1a). We have added the corresponding raw data in the Supplementary raw data Excel file.

g. supplementary videos of raw data used to create Figure 6a-f,

Two videos are now provided:

Supplementary Video File 4: Tether dynamics of *Mical1^{+/+}* platelets.

Supplementary Video File 5: Tether dynamics of *Mical1^{-/-}* platelets.

As described above, the experiments Figure 6b-f were not monitored in real time. We have added the corresponding raw data in the Supplementary raw data Excel file.

h. supplementary videos of raw data used to create Figure 7e, 7h

Six videos are now provided:

Supplementary Video File 6: Rolling experiment of *Mical1^{+/+}* platelets.

Supplementary Video File 7: Rolling experiment of *Mical1^{-/-}* platelets.

Supplementary Video File 8: Rolling experiment of *Mical1^{+/+}* platelets treated with DMSO.

Supplementary Video File 9: Rolling experiment of *Mical1^{+/+}* platelets treated with LatA.

Supplementary Video File 10: Rolling experiment of *Mical1^{-/-}* platelets treated with DMSO.

Supplementary Video File 11: Rolling experiment of *Mical1^{-/-}* platelets treated with a low dose of LatA (rescue experiment).

i. Raw data of thrombus size after stability challenge in Figure 4b and Figure 5b (in addition to the existing quantification of thrombus instability)

These data are now included in the Supplementary raw data Excel file.

6. Data presentation. Figure 7i and j use green and red immunofluorescence images and an overlay. Given that red/green colorblindness is the most common type of colorblindness, this reviewer suggests that the authors choose different colors. There are many alternative colorblind-friendly options.

We changed with colorblind-friendly options (cyan/red).

7. Typos, grammar, and clarifications.

a. The capitalization of MICAL1 is inconsistent. The manuscript uses both “MICAL1” and “Mical1”.

We apologize for the mistake, and we have replaced Mical1 with MICAL1 when referring to the protein.

b. Supplementary table 2 is titled “.... Mice separated by gender.” “Male” and “female” are terms to describe sex, not gender. The term “gender” should be changed to “sex”.

We have now changed “gender” to “sex”.

c. Page 7, line 27-28 states, “we therefore hypothesized that this embolism could be due to impaired adhesion...” Given that the authors showed that the KO has more platelet adhesion under low shear, the reviewers suggest that perhaps the authors mean, ““we therefore hypothesized that this embolism could be due to impaired adhesion in high shear conditions...”

As suggested, we have added “in high shear conditions”.

d. This reviewer encourages the authors to expand upon the translational implications of Mical1 in patient care.

We have now expanded upon the translational implications of MICAL1 in patient care in the discussion (p. 12)

“These results suggest that MICAL1 could represent a promising therapeutic target for pathologies characterized by VWF hyperactivation, such as in patients who have undergone transcatheter aortic valve implantation (TAVI) or those with type 2B von Willebrand disease. Furthermore, from a clinical perspective, the existence of a *MICAL1* variant associated with bleeding suggests that the F-actin limitation by MICAL1 plays a role in human platelet function⁵⁷.”

Reviewer #4 (Remarks to the Author):

Reviewer #5 (Remarks to the Author):
